# BaRISTA: Brain Scale Informed Spatiotemporal Representation of Human Intracranial Neural Activity

**Lucine L. Oganesian**[†1]   **Saba Hashemi**[†2]   **Maryam M. Shanechi** [∗1−4]

University of Southern California, Los Angeles, CA

{loganesi,saba.hashemi,shanechi}@usc.edu

## Abstract

Intracranial recordings have opened a unique opportunity to simultaneously measure activity across multiregional networks in the human brain. Recent works have focused on developing transformer-based neurofoundation models of such recordings that can generalize across subjects and datasets. However, these recordings exhibit highly complex spatiotemporal interactions across diverse spatial scales, from the single-channel scale to the scale of brain regions. As such, there remain critical open questions regarding how best to encode spatial information and how to design self-supervision tasks that enable the learning of brain network patterns and enhance downstream decoding performance using such high-dimensional, multiregional recordings. To allow for exploring these questions, we propose a new spatiotemporal transformer model of multiregional neural activity and a corresponding self-supervised masked latent reconstruction task, designed to enable flexibility in the spatial scale used for token encoding and masking. Applying this model on publicly available multiregional intracranial electrophysiology (iEEG) data, we demonstrate that adjusting the spatial scale for both token encoding and masked reconstruction significantly impacts downstream decoding. Further, we find that spatial encoding at larger scales than channel-level encoding, which is commonly used in existing iEEG transformer models, improves downstream decoding performance. Finally, we demonstrate that our method allows for region-level token encoding while also maintaining accurate channel-level neural reconstruction. Taken together, our modeling framework enables exploration of the spatial scales used for token encoding and masking, reveals their importance towards self-supervised pretraining of neurofoundation models of multiregional human brain activity, and enhances downstream decoding performance.

## 1   Introduction

Intracranial electroencephalography (iEEG) provides a direct window into the human brain by enabling the simultaneous recording of high-dimensional neural activity across multiple brain regions, thus measuring diverse spatial scales from single channels to large-scale brain networks. Enabling the modeling of such recordings can provide a unique opportunity to study functional brain networks associated with complex behavioral and cognitive processes [1–4] and develop translational technologies such as brain-computer interfaces [5, 6]. Compared with non-invasive approaches such as fMRI

---

∗Corresponding author: shanechi@usc.edu.

† Equal contribution.

[1]Ming Hsieh Department of Electrical and Computer Engineering, University of Southern California

[2]Thomas Lord Department of Computer Science, University of Southern California

[3]Alfred E. Mann Department of Biomedical Engineering, University of Southern California

[4]Neuroscience Graduate Program, University of Southern California

39th Conference on Neural Information Processing Systems (NeurIPS 2025).

or scalp EEG, iEEG yields a more direct measurement of brain activity with rich temporal dynamics. Furthermore, while intracortical recordings of spiking activity typically focus on measuring neuronal populations within a local brain circuit or region (e.g., motor cortex), iEEG data is typically collected from sparsely-placed electrodes across much larger spatial scales of several brain regions at once. As such, modeling of iEEG presents distinct challenges due to its complex spatiotemporal structure. Towards the goal of learning rich spatiotemporal representations for iEEG activity, there has been keen interest in developing iEEG neurofoundation models that can generalize across different subjects and datasets, paralleling recent efforts for spiking data [7–10], non-invasive EEG [11], and fMRI [12, 13]. To do so, recent works have leveraged large transformer-based models, often pretrained with self-supervision, to learn rich representations of human iEEG data with demonstrated efficacy in downstream tasks and cross-subject generalization [14–19]. Despite the progress that has been made, there still remain critical open questions on how best to incorporate spatial information when designing and training such models.

First, while prior works have largely used standard positional encoding methods for providing temporal information to the transformer model (e.g., sine-cosine, rotary, learnable), there still exists no unified approach for encoding space during neural tokenization - here defined as the process of transforming continuous neural recordings into finite-dimensional input tokens for the transformer encoder. Previous approaches have either not encoded space [14], collapsed it across prespecified channels chosen based on neuroscientific knowledge [17], or encoded space but at the scale of single channels [16, 18, 19]. As such, developing models that enable a larger than channel-level spatial scale for token encoding and studying the effect of such larger-scale encoding remains unexplored. Second, it is not clear if and how spatial information should be incorporated into self-supervised model pretraining. Prior works have pretrained spatiotemporal models of iEEG activity, using either supervised [18] or self-supervised methods [14–16, 19], and demonstrated transferability across tasks, subjects, or sessions. Among these, one approach has used a discrimination task to identify if a channel had been randomly replaced in an ensemble of channels [19]. However, a critical remaining question is how different spatial scales would impact self-supervised pretraining. Indeed, within the context of masked pretraining, none of the existing approaches have explicitly incorporated the notion of space within their masking strategy and have instead typically selected random channels to mask and reconstruct. Thus, it remains unclear if channel-based masking is preferred over larger scales of masking, such as brain region-based, when modeling multiregional neural activity.

**Contributions**   Here we address the above challenges by developing a neural tokenization and spatial encoding scheme that maintains individual channel temporal statistics while also enabling spatial encoding at larger spatial scales. Further, to study the impact of spatial scales on model pretraining with a self-supervised masked reconstruction task, we also develop an end-to-end training procedure that trains a model to reconstruct targets that are masked based on spatial meta-information, supporting masking both at the single channel scale as well as larger brain region scales. We call our modeling framework BaRISTA. In summary, our contributions are the following:

1. We develop a spatiotemporal transformer model of intracranial neural activity and an associated masked latent reconstruction pretraining task. Within our framework, we dissociate the selection of spatial encoding from spatial masking to isolate the effects of one from the other on learned representations and overall pretrained model performance.

2. Using our framework, we investigate the impact of spatial resolution on model pretraining and downstream task performance, observing that spatial encoding at larger spatial scales improves downstream decoding performance over channel-level encoding.

3. We demonstrate with a downstream masked channel reconstruction task that our modeling and pretraining approach is able to incorporate larger-scale spatial information without sacrificing knowledge of individual channel temporal statistics.

## 2   Related Work

**Spatiotemporal models of intracranial neural activity**   Several prior works have proposed spatiotemporal models of iEEG activity using different approaches for encoding spatial information. Brant and its subsequent iterations did not explicitly encode the spatial axis and only utilized standard positional encoding [20] of the temporal axis in their models [15, 16]. Zheng et al. [17] proposed an

approach to model iEEG activity within preselected brain regions by pooling all channels within a region, collapsing out the spatial axis, and thereby precluding the need for explicit spatial encoding. Finally, both Mentzelopoulos et al. [18] and Chau et al. [19] encoded space at the single-channel scale by incorporating neuroanatomical information and utilizing each channel's volumetric 3D coordinate to construct the corresponding token's spatial encoding vector. However, to our knowledge, no prior work on modeling iEEG data has looked at maintaining channel-level tokens while encoding spatial information at larger spatial scales (as we do here), such as the brain regions in which the channels are located.

**Self-supervised masked modeling of neural data** Paralleling demonstrations in population spiking [7, 9], fMRI [12, 13], and EEG [11] neurofoundation models, there have been recent efforts showing the utility of masked self-supervised pretraining for spatiotemporal models of iEEG data [15–17]. Most of these prior works have typically used a random masking strategy at the level of individual channels, following standard procedure in other domains such as vision [21, 22] and language [23]. However, a random channel-level masking strategy for target selection may not necessarily be the most effective for iEEG data due to the unique statistical properties and functional roles associated with spatially distributed channel recordings. As such, here we develop a masked model pretraining task that allows for flexible specification of masking targets based on user-specified meta-information, for example domain knowledge of neuroanatomy or that of functional brain network activity. This allows us to explore masking both at the single-channel scale and at larger scales. Finally, we further differentiate from prior approaches by training our model, which consists of a neural tokenizer and a combined spatiotemporal encoder, end-to-end to perform reconstruction in the latent rather than observation space.

## 3 Methods

To assess the impact that spatial encoding and masking each have on representation learning and downstream model performance, we developed a new spatiotemporal transformer model and a corresponding pretraining framework that allowed us to independently adjust the spatial scales used for token encoding and target selection in the masked reconstruction task. We first describe how we chose the spatial scales we tested and how we flexibly incorporated that spatial information into our framework. We then present our transformer model architecture and our self-supervised pretraining procedure. Finally, we discuss our evaluation schemes.

### 3.1 Spatial scales investigated

Intracranial neural recordings are multivariate time-series collected from electrode channels that span multiple brain regions. As such, there is an inherent notion of space, both at small (e.g., 3D channel coordinates) and large (e.g., brain regions) spatial scales. Here, we explore the choice of spatial scale within the context of masked self-supervised pretraining. For our investigation, we choose three spatial scales based on neuroanatomical meta-information to test (see Figure 1 and Appendix D for details):

(1) **Channel** Similar to [18] and [19], channel $(x, y, z)$ (left, posterior, and inferior, LPI [24]) coordinates in MRI volumetric space are used for spatial token encoding. In this regime, channels are randomly selected and masked.

(2) **Atlas parcellations** Spatial encoding and masking is based on electrode localization and channel assignments to cortical parcellations using standard brain atlases (e.g., Destrieux or Desikan-Killiany atlases in Freesurfer). Here we choose to use parcel assignments based on the Destrieux [25] atlas as it contains more parcels and therefore permits finer-grained analyses. We also additionally include subcortical structures (e.g., hippocampus) that were annotated in the provided dataset (Section 4.1).

(3) **Lobes** Spatial encoding and masking is performed at scales corresponding to brain lobes. We also include regions that are not considered lobes (e.g., cingulate), but are often regions of interest across various neuroscience domains. Lobe identities for each channel are designated based on the Desikan-Killiany atlas as per the appendix of [26].

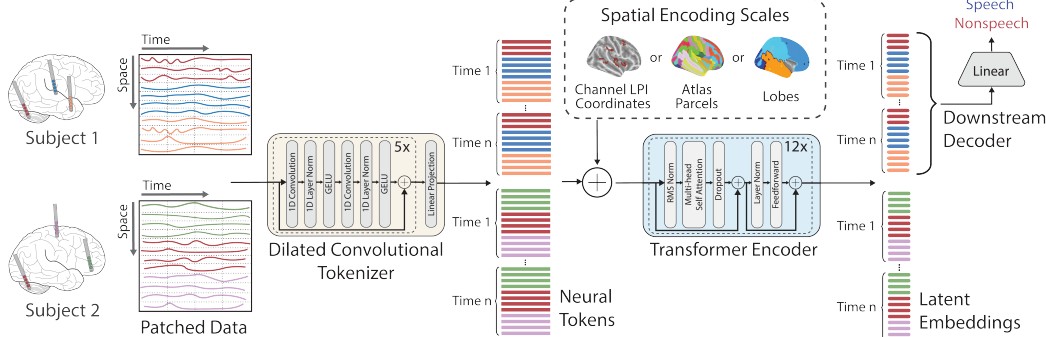

Figure 1: **BaRISTA model architecture.** Subject data is first channel-wise patched along the time axis and encoded using a tokenizer (dilated convolutional temporal encoder and linear projection layer). Then, spatial information is encoded based on a prespecified spatial scale; here we explore channel-level volumetric LPI coordinates, atlas parcels, and lobes. Neural tokens are passed as inputs to the encoder transformer, which provides the embeddings used for downstream tasks.

## 3.2 Model architecture

Our model architecture and tokenization scheme are shown in Figure 1. Given a multivariate time-series of neural activity $\mathbf{X} \in \mathbb{R}^{C \times T}$, where $C$ denotes the number of recording channels and $T$ denotes time, we first tokenize channels as univariate signals (i.e., agnostic to space), following common practice [14, 15, 18, 27, 28]. We create temporal patches of each channel that are of length $L$ (e.g., 250 milliseconds), such that $\mathbf{P}_{ij} \in \mathbb{R}^L$ indicates the $i$-th patch of length $L$ for the $j$-th channel. Our tokenizer, denoted by $\mathcal{F}$, consists of a temporal encoder and a linear projection layer. In the first step of tokenization, each temporal patch is passed through a shared temporal encoder. In practice this encoder can take any form; here we choose a dilated convolutional neural network (CNN) [29–32], both to account for the input signal's continuous nature and because of prior domain knowledge about the importance of oscillatory features in neural activity [1, 33]. Next, we apply a linear layer on the output of the temporal encoder to create tokens of dimension $d$, such that $\mathbf{B}_{ij} = \mathcal{F}(\mathbf{P}_{ij}) \in \mathbb{R}^d$ denotes the token corresponding to patch $\mathbf{P}_{ij}$.

To encode space we add a learnable embedding vector, denoted by $\mathbf{E}_j := e^{\mathrm{sp}(j)} \in \mathbb{R}^d$, that corresponds to the $j$-th channel's spatial category, $\mathrm{sp}(j)$. Note, this category depends on the selected scale among the three spatial scales explored here (Section 3.1) and refers to the channel's spatial designation within the selected scale. At larger scales, two channels may have the same spatial encoding if they belong to the same category (e.g., the same parcel assignment). The number of unique categories within a given spatial scale determines the size $|\mathcal{K}|$ of the learnable spatial embedding dictionary for that scale (more details about spatial categories are provided in Appendix D). Using the spatially-encoded tokens, denoted as $\mathbf{S}_{ij} = \mathbf{B}_{ij} + \mathbf{E}_j$, we create the transformer input token sequences of length $nC$, where $n$ indicates the number of temporal patches for one channel. Specifically, we order all channels' tokens within an input sequence as

$$\mathbf{S} = [\mathbf{S}_{11}, \mathbf{S}_{12}, \cdots, \mathbf{S}_{1C}, \mathbf{S}_{21}, \cdots, \mathbf{S}_{nC}] \in \mathbb{R}^{(nC) \times d},$$

such that temporal and spatial information are interleaved. This allows us to have a single encoder transformer that can attend to space and time concurrently – unlike some prior work that cascaded the temporal and spatial transformers [15, 18, 19]. We also note that because transformer input sequences can be of variable lengths $C$ and $n$ here are also not fixed, meaning our method can support modeling sessions with differing channel and patch counts. To encode temporal information at the token level, we use rotary positional embeddings (RoPE) in our transformer's attention layers [34]. Finally, the outputs of our spatiotemporal encoder transformer model, $\mathbf{Z} = \mathcal{G}(\mathbf{S}) \in \mathbb{R}^{(nC) \times d}$, are used as the neural embeddings for all downstream tasks (Section 3.4). In Appendix J (Tables 14 and 15), we present ablation results on our choice of temporal encoder (CNN) and the combined attention module. Comprehensive details on model architecture are provided in Appendix B.

## 3.3 Spatially masked latent reconstruction

Our training procedure is shown in Figure 2. We train BaRISTA using a self-supervised masked token reconstruction task, which differs from prior work in two ways. First, we use the selected spatial

scale to guide masking, rather than only masking randomly-selected channels [15, 16] or tokens [17]. Second, unlike some prior iEEG models [17, 19], we simultaneously train both the tokenizer and encoder transformer to perform masked reconstruction in the latent token space.

During training, we randomly select a subset of spatial categories within the input data to mask, denoted by $SP_{target}$. We use all the tokens that correspond to the selected spatial categories as our target tokens, $\mathbf{B}_{\text{target}}$, such that

$$\mathbf{B}_{\text{target}} = \{\mathbf{B}_{ij}\}_{\text{sp}(j) \in SP_{\text{target}}}.$$

We note that the selection of target spatial categories is constrained such that the total number of masked tokens $|\mathbf{B}_{\text{target}}|$ corresponds to our desired masking percentage – a hyperparameter of our model. All remaining tokens are used as observation tokens, $\mathbf{B}_{\text{obs}}$. While observation tokens are obtained using our original tokenizer (top row, Figure 2A), we use a separate target tokenizer $\tilde{\mathcal{F}}$ for the target tokens (bottom row, Figure 2A). The target tokenizer is updated with an exponential moving average (EMA) of the original tokenizer weights. In our online network (top row Figure 2A), the target tokens are replaced with a shared learnable mask token, $\mathbf{M}$. The spatial encoding for each token is added to the masked input sequence as described in Section 3.2. So, for example, if $SP_{target}$ contains channels 1 and 2, then the input sequence

$$\mathbf{S} = [\mathbf{S}_{11}, \mathbf{S}_{12}, \mathbf{S}_{13}, \cdots, \mathbf{S}_{1C}, \mathbf{S}_{21}, \mathbf{S}_{22}, \cdots, \mathbf{S}_{nC}]$$

would become the masked input sequence

$$\mathbf{S}_{\text{masked}} = [\mathbf{M} + \mathbf{E}_1, \mathbf{M} + \mathbf{E}_2, \mathbf{S}_{13}, \cdots, \mathbf{S}_{1C}, \mathbf{M} + \mathbf{E}_1, \mathbf{M} + \mathbf{E}_2, \cdots, \mathbf{S}_{nC}] \in \mathbb{R}^{(nC) \times d}$$

where $\mathbf{E}_j$ denotes the spatial encoding for the corresponding $j$-th masked token. Temporal position for masked tokens was encoded using RoPE, as in Section 3.2.

After obtaining the latent embeddings $\mathbf{Z}$ from the transformer, we pass the embeddings for the masked tokens to a predictor network, $\mathcal{H}$, to perform target token reconstruction (Figure 2A). Here, we use a multi-layer fully-connected network (MLP) as our predictor, $\mathcal{H}$. Our training loss is the average mean-squared error between predicted tokens, $\hat{\mathbf{B}}_{ij} = \mathcal{H}(\mathcal{G}(\mathbf{M} + \mathbf{E}_j | \mathbf{S}_{\text{masked}}))$, and target tokens, $\tilde{\mathbf{B}}_{ij} = \tilde{\mathcal{F}}(\mathbf{P}_{ij})$:

$$\mathcal{L} = \frac{1}{|\mathbf{B}_{\text{target}}|} \sum_{i \in \{1..n\}, j \in SP_{\text{target}}} \|\tilde{\mathbf{B}}_{ij} - \hat{\mathbf{B}}_{ij}\|_2^2,$$

For all downstream tasks, we use the EMA-updated target tokenizer with the transformer backbone as our pretrained model. For our channel reconstruction downstream task described in Section 3.4 (Figure 2B), we retain the trained predictor network $\mathcal{H}$. Additional details on model training are provided in Appendix C.

### 3.4  Downstream evaluation

We evaluate the validity of our training procedure and the effectiveness of our learned model using several downstream tasks. We also evaluate the impact of spatial scale, both for token encoding and masking, on the same tasks. To do so, we first validate our pretrained model's performance on two language-related downstream tasks used in [14, 19]: classification of speech vs. non-speech audio and identification of words that correspond to sentence onsets. Classification performance is reported as an average across all hold-out test sessions for 5 finetuning seeds each (see Section 4.1 and Appendix A). As baselines, we compare our pretrained model's finetuned performance against a finetuned, randomly-initialized version of itself and two state-of-the-art (SOTA) spatiotemporal iEEG models: Population Transformer (PopT) [19] and Brant [15].

Second, we use the flexibility afforded by our framework to pretrain BaRISTA with different spatial token encoding and masking scales, and we compare the different configurations based on their performance on the language-related downstream classification tasks. Third, we also evaluate our pretrained model's finetuned performance on *masked* neural reconstruction in the *observation* space as another downstream task. We finetune models pretrained with different spatial configurations using a mean-squared error reconstruction loss computed on an individual channel basis. Distinct from the language-related classification tasks above, here we also finetune the prediction head $\mathcal{H}$ from our pretraining task to perform masked channel reconstruction (Figure 2). Details on the classification tasks and the reconstruction task setup, including the training procedure we had to develop to teach the model to reconstruct channel activity from masked tokens, are provided in Appendices E.2 and E.3, respectively.

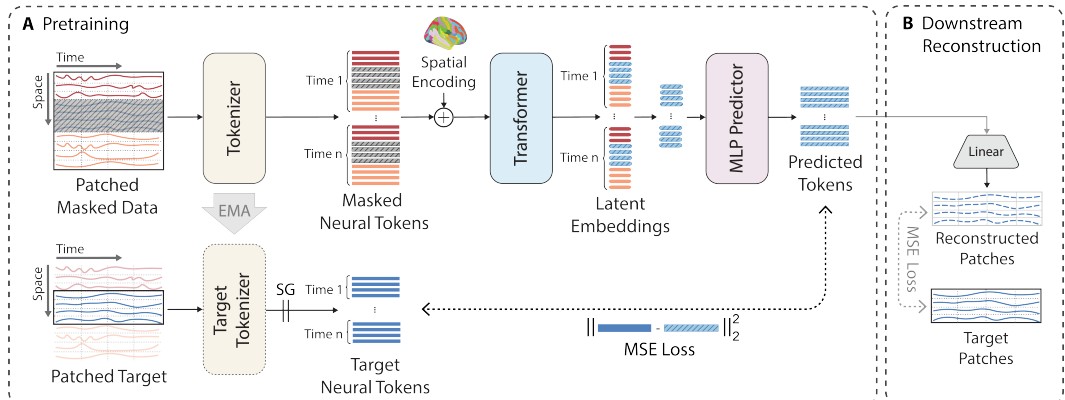

Figure 2: **BaRISTA is pretrained with a masked latent token reconstruction task. A.** We randomly select observed and target spatial categories, which are encoded with an online (top) and target (bottom) tokenizer, respectively. Target tokens are replaced with a learnable mask token before being embedded by the transformer (top). The embeddings for the masked tokens are used to predict the target tokens as per a mean-squared error loss. The trained encoder transformer and target tokenizer are used for downstream tasks. SG=stop gradient. **B.** We use a linear layer to reconstruct raw channel time-series activity from masked target tokens, using the predictions provided by the pretrained predictor network. A mean-squared error loss between true and reconstructed neural activity is used for finetuning.

## 4 Experimental results

### 4.1 Dataset and evaluation methods

For our experiments we used the publicly available Brain Treebank dataset [35], which consists of intracranial recordings from 10 epilepsy patients collected over a total of 26 sessions as they watched Hollywood films. Film transcripts that are aligned to neural activity are also provided. The iEEG recordings cover multiple brain regions across both hemispheres, including the temporal and frontal lobes, which are known to support auditory and language processing. Neural data is provided at a sampling rate of 2048 Hz. We followed similar preprocessing procedures on raw data (e.g., filtering) as outlined in [14, 19, 35] but generated our downstream data segments differently in two ways to enable two sets of evaluations (details are in Appendices A and K). For our main evaluation, we generated non-overlapping 3-second-long neural data segments and randomly assigned them to 80/10/10 train/valid/test splits; we present the results of this analysis in Sections 4.2 and 4.3. However, since enforcing no overlap requires dropping some of the labeled segments, we also performed an alternative evaluation that let us use more of the annotations provided by the Brain Treebank dataset [35] for downstream training. In this evaluation, we allowed for overlapping neural segments and generated the 80/10/10 train/valid/test splits chronologically in time to avoid any overlap between these splits. This procedure increased the amount of labeled data and additionally enabled evaluation on 2 more downstream tasks. We provide the results of the second evaluation in Appendix K. Our findings across both evaluation schemes were consistent, thus providing a rigorous validation of our conclusions. Finally, to further validate our framework and our baseline comparisons, we confirmed that we were able to reproduce the PopT downstream classification results reported in [19] when using their original downstream segments (see Appendix E.1). For all downstream classification tasks we report the average performance (+/- standard error of measure, s.e.m.) over the 7 test hold-out sessions, with 5 finetuning seeds for each task.

Lastly, for pretraining, we generated 3-second-long non-overlapping neural segments which we separated into 80/10/10 train/valid/test data splits. We pretrain on 17 of the sessions and hold-out 2 and 7 sessions for validation and test, respectively [14, 19].

### 4.2 BaRISTA's flexible spatial encoding enables decoding improvements over baselines

In Table 1 we report the average classification ROC-AUC over all test sessions and finetuning seeds ($n = 35$ points total). Our results show that our model outperforms all alternative models by enabling flexibility over spatial encoding. First, our pretraining improves downstream performance

Table 1: Classification results (mean AUC $\pm$ s.e.m.). Within each task, asterisk* indicates the best-performing (**bolded**) model is significantly better than second-best (underlined) model with p-value <1e-5 (Wilcoxon signed-rank test).

| Model | Sentence Onset | Speech/Non-Speech |
|---|---|---|
| Brant [15] | $0.767 \pm 0.017$ | $0.691 \pm 0.017$ |
| PopT+Brainbert [19] | $\underline{0.795 \pm 0.014}$ | $\underline{0.775 \pm 0.016}$ |
| BaRISTA (channels/channels) | $0.778 \pm 0.019$ | $0.764 \pm 0.020$ |
| BaRISTA (parcels/channels) | $\mathbf{0.862 \pm 0.016}^{*}$ | $\mathbf{0.869 \pm 0.016}^{*}$ |
| BaRISTA (random initialization, channels) | $0.688 \pm 0.017$ | $0.616 \pm 0.019$ |
| BaRISTA (random initialization, parcels) | $0.683 \pm 0.017$ | $0.627 \pm 0.018$ |

Table 2: Downstream classification results of different spatial encoding/masking configurations (mean AUC +/- s.e.m.). Best results in **bold**.

| | Mask Encode | Channels | Parcels | Lobes | Random Init. |
|---|---|---|---|---|---|
| Sentence Onset | **Channels** | $0.778 \pm 0.019$ | $0.710 \pm 0.017$ | $0.654 \pm 0.019$ | $0.688 \pm 0.017$ |
| | **Parcels** | $\mathbf{0.862 \pm 0.016}$ | $\mathbf{0.861 \pm 0.014}$ | $0.841 \pm 0.015$ | $0.683 \pm 0.017$ |
| | **Lobes** | $0.842 \pm 0.016$ | $0.816 \pm 0.017$ | $0.840 \pm 0.014$ | $0.681 \pm 0.017$ |
| Speech | **Channels** | $0.764 \pm 0.020$ | $0.668 \pm 0.015$ | $0.652 \pm 0.017$ | $0.616 \pm 0.019$ |
| | **Parcels** | $\mathbf{0.869 \pm 0.016}$ | $\mathbf{0.866 \pm 0.015}$ | $0.845 \pm 0.017$ | $0.627 \pm 0.018$ |
| | **Lobes** | $0.841 \pm 0.019$ | $0.823 \pm 0.015$ | $0.840 \pm 0.015$ | $0.628 \pm 0.017$ |

compared to randomly initialized versions of our model. Moreover, pretraining using channel-level encoding and masking yields performance roughly on par with recent iEEG models, both of which use channel-level encoding (none of the differences between our channel-level model and baselines were significant, except for our model being significantly better than Brant for the speech task, Wilcoxon signed-rank p-value 3.869e-05). Interestingly, however, when using larger-scale parcel-level encoding and channel-level masking, our model achieves higher overall downstream performance compared to these SOTA iEEG models (difference with PopT significant with Wilcoxon signed-rank p-values 5.014e-06 and 2.328e-10 on sentence onset and speech tasks, respectively). Overall, the results in Table 1 demonstrate that by affording flexibility over the spatial encoding scale during masked reconstruction pretraining, our model can improve downstream task performance. For individual subject performance we refer readers to Appendix Table 11. Similar results held in our second evaluation with chronological splits (see Appendix Table 16).

## 4.3 Larger scale spatial encoding enhances downstream performance

Next, we investigated the impact of spatial scale in both token encoding and masking and used our framework's flexibility to dissociate these two effects. To do so, we pretrained our model using 9 distinct spatial encoding/masking combinations with the 3 different spatial scales described in Section 3.1, and evaluated each pretrained model's performance on the same language-related tasks in Table 1. We present both finetuned and random initialization results in Table 2; we note that encoding/masking combinations presented in Table 1 are subcomponents of the complete results presented in Table 2.

First, we find that the choice of spatial scale has a significant impact on the performance of the pretrained model (Table 2 and Figure 3). Second, we see that the choice of spatial encoding, rather than spatial masking, has a larger impact on final downstream performance for both tasks. Third, interestingly, we find that channel-level encoding underperforms larger spatial scale encodings regardless of the spatial masking scale. To further isolate and quantify the sources of variability, we performed a two-way ANOVA [36] with spatial encoding and spatial masking as the independent variables and the ROC-AUC values as the dependent variable; we Bonferroni correct p-values to account for tested conditions (e.g., two downstream tasks, etc.). The two-way ANOVA revealed that both independent variables had a statistically significant effect on downstream task performance with no significant interaction (sentence onset: encoding $p < 1e-3$, masking $p = 0.010$; speech: encoding $p < 1e-3$, masking $p = 0.037$). As another observation, by using BaRISTA's flexibility in designating encoding and masking spatial scales, we found that when using channel-level encoding,

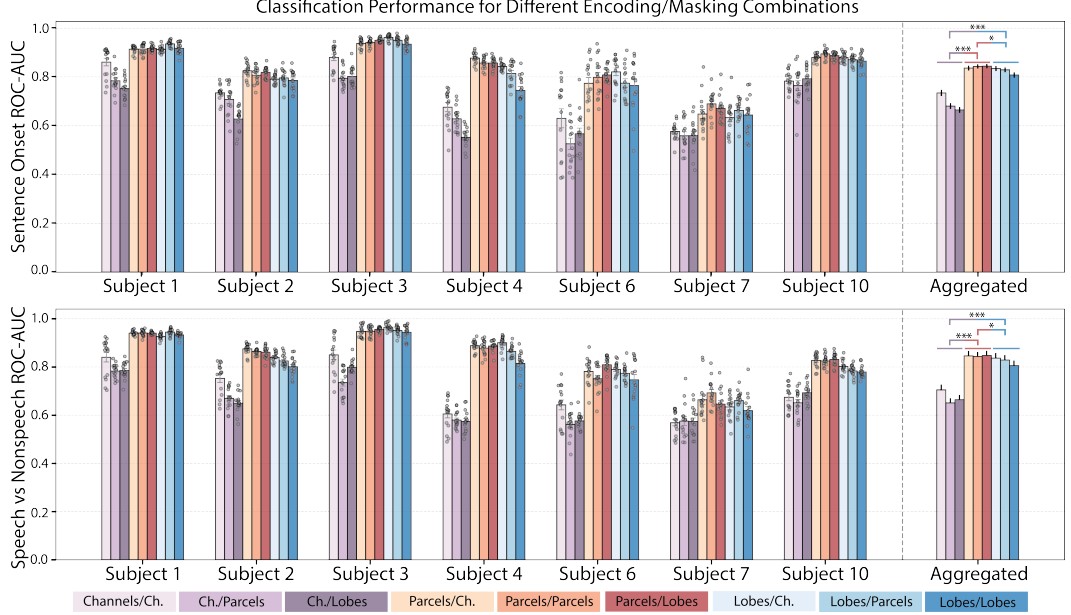

Figure 3: **Channel-level spatial encoding underperforms parcel- and lobe-level encoding across all subjects for both downstream tasks, suggesting the importance of larger spatial scales in masked reconstruction pretraining.** Scatter points correspond to individual trials (3 pretraining and 5 finetuning seeds), error bars correspond to s.e.m. Aggregated results pool trials across all subject sessions for each condition. Two-sided Wilcoxon signed-rank tests were conducted between spatial encoding pairs, with $*$ and $***$ indicating p-values $\in [1e-5, 1e-10]$ and $\leq 1e-15$, respectively. Ch.=channels.

channel-level masking works better than masking at larger scales, which may be an important consideration if a given application requires channel-level encoding. Furthermore, we note that the choice of spatial encoding has no impact in the randomly-initialized setting. Per-subject results are presented in Figure 3. Here and in Section 4.2, we present results for a single pretraining seed per spatial encoding/masking category that was selected based on validation hold-out performance in the two downstream language tasks; we do this to be consistent with prior works that presented results for a single pretraining seed (e.g., [15, 19]). We also present downstream classification results averaged across 3 different pretraining seeds, in addition to the 5 finetuning seeds, in Appendix F (Table 10). We find similar results in our second evaluation with chronological splits (see Appendix Table 17).

In summary, our results show that larger than channel-level spatial scales, particularly for neural token encoding, can critically improve downstream classification performance, demonstrating that the choice of spatial scale can be important in self-supervised masked reconstruction pretraining. Additional model interpretability results are presented in Appendix G (Figures 7 and 8).

### 4.4 BaRISTA can maintain channel-level reconstruction with larger-scale spatial encoding

Beyond looking at higher-order language-related tasks, we also considered pretrained model performance on a masked channel reconstruction task in the observation space. We first used the same setup as our pretraining task to predict the target tokens from the masked tokens, using our pretrained model and the pretrained predictor network $\mathcal{H}$ (Figure 2A). To reconstruct the target channel's raw time-series activity from the predicted neural tokens, we added a linear head after the predictor network, $\mathcal{H}$, that maps the predicted tokens, $\hat{\mathbf{B}}_{ij} = \mathcal{H}(\mathcal{G}(\mathbf{M} + \mathbf{E}_j | \mathbf{S}_{\text{masked}}))$, to the corresponding raw time-series patch, $\hat{\mathbf{P}}_{ij}$ (Figure 2B). During evaluation, we mask out one channel at a time and report the average reconstruction mean-squared error (MSE) and coefficient of determination ($R^2$) across all masked channels for the 7 held-out test sessions (Section 4.1) in Table 3. As a baseline, we include the performance of randomly-initialized models that use the same spatial encoding. Interestingly, we see that finetuned models using parcel-level spatial encoding are able to achieve reconstruction performance comparable to finetuned channel-level encoded models. This suggests that the framework is capable of modeling larger than channel-level spatial interactions without loss of individual channel

information. For further illustration, we show example reconstruction traces for 2 of our pretrained models (channel/channel and parcel/parcel) in Figure 4. We can observe qualitatively that our method more accurately reconstructs low-frequency vs. high-frequency content. We quantitatively confirm this observation by performing a spectral analysis of the reconstruction results in Appendix I. Finally, full experimental details are provided in Appendix E.3, and subject-specific performance is provided in Appendix H (Table 12).

Table 3: Masked channel reconstruction performance (mean $\pm$ s.e.m.). Best results in **bold**, second-best results underlined. Init=initialization.

| | Encode \ Mask | Channels | Parcels | Lobes | Random Init. |
|---|---|---|---|---|---|
| **MSE ↓** | **Channels** | $0.397 \pm 0.040$ | $\mathbf{0.354 \pm 0.032}$ | $0.478 \pm 0.036$ | $0.566 \pm 0.028$ |
| | **Parcels** | $0.391 \pm 0.019$ | $0.413 \pm 0.023$ | $0.417 \pm 0.027$ | $0.846 \pm 0.028$ |
| | **Lobes** | $0.753 \pm 0.039$ | $0.951 \pm 0.014$ | $0.853 \pm 0.029$ | $0.965 \pm 0.022$ |
| **$R^2$ ↑** | **Channels** | $0.603 \pm 0.040$ | $\mathbf{0.646 \pm 0.032}$ | $0.522 \pm 0.036$ | $0.434 \pm 0.028$ |
| | **Parcels** | $0.609 \pm 0.019$ | $0.587 \pm 0.023$ | $0.583 \pm 0.027$ | $0.155 \pm 0.028$ |
| | **Lobes** | $0.247 \pm 0.039$ | $0.049 \pm 0.014$ | $0.147 \pm 0.029$ | $0.035 \pm 0.022$ |

Figure 4: **Example reconstruction traces from masked tokens using models pretrained with different spatial encoding/masking pairs for two different 3-second segments.** Parcel-level spatial encoding performs comparably to channel-level encoding in channel reconstruction performance, suggesting that channel-specific information is not lost when modeling with larger spatial scales. For visualization purposes, raw reconstruction outputs have been smoothed using SciPy's [37] Savitzky-Golay filter with a window size of 5 and polynomial order 2.

## 4.5 Pretrained BaRISTA generalizes to unseen subjects and scales with pretraining data

To assess the ability of BaRISTA to generalize to completely unseen subjects, we conducted an analysis using our downstream language tasks in which we held-out *all* sessions for a test subject during pretraining and evaluated the resulting model's classification performance for the unseen subject. We performed this analysis for each of the test subjects specified in Appendix Table 5 and used the parcel/channels model configuration reported in Table 1. Average results are presented in Table 4 and individual subject results are presented in Appendix Table 11. While minor performance degradation is seen, as expected, the performance on unseen subjects is still higher than the two SOTA baselines and our randomly initialized models (Table 1). We also compared the downstream classification performance of the same parcels/channels BaRISTA model when pretrained using 5%, 10%, 25%, 50%, and 75% of the total available pretraining data. Doing so, we observed that our model's downstream performance on the same downstream language tasks successfully scaled with more pretraining data (Figure 5). To get the desired percentage, we added sessions randomly one by one, such that their total number of segments matches the desired data percentage. To ensure the results were not biased by a specific sampling order, we repeated this process with 5 different random seeds. We also adjusted the number of epochs for pretraining when using a lower percentage of data, such that the total number of parameter updates for each of the data size percentages was roughly comparable. We find similar patterns of generalizability and scaling using our second evaluation with chronological splits, provided in Appendix K.1 (Table 19 and Figure 9).

## 5 Discussion and future directions

There are several interesting directions for future work that may further improve our modeling framework. First, although here we defined our spatial scales based on anatomical designations, our

Table 4: Generalizability to new subjects: downstream results of our parcels/channels model for both standard pretraining and pretraining with the target subject completely held-out (mean +/- s.e.m.). Results are averaged across 5 finetuning seeds.

| Model | Sentence Onset | Speech/Non-Speech |
|---|---|---|
| BaRISTA (parcels/channels, Held-out) | $0.841 \pm 0.016$ | $0.852 \pm 0.013$ |
| BaRISTA (parcels/channels, Included) | $0.862 \pm 0.015$ | $0.869 \pm 0.016$ |

model is flexible in terms of what "spatial" definitions to use. As such, alternative definitions, for example based on the functional roles of brain regions regardless of their anatomical designation [13] can also be utilized within our framework. Indeed, in future work it may be interesting to use the flexibility of spatial encoding enabled by our framework for hypothesis-driven testing of encoding scales on a variety of downstream tasks that exhibit different degrees of complexity, including simpler sensory tasks. Doing so may yield further improvements in model performance and/or insights about the encoding of various behavioral and cognitive states.

Second, in all of our experiments we used spatial-only masking in order to study the impact of spatial scales on model pretraining. Future work can explore integrating more diverse masking procedures [9, 13], such as masking across space and time, to further improve overall model performance and to potentially help facilitate learning of richer representations of iEEG recordings. Finally, we used a dilated CNN for temporal encoding and saw that our modeling framework, even when using larger than channel-level scale, was able to maintain channel-level temporal statistics to perform reconstruction (Sections 3.3 and 4.4). Nevertheless, exploring alternative temporal encoding schemes, such as temporal pyramid pooling [38] or a combination of short-term and long-term encoders [39], may further improve channel reconstruction and will be interesting to explore in the future.

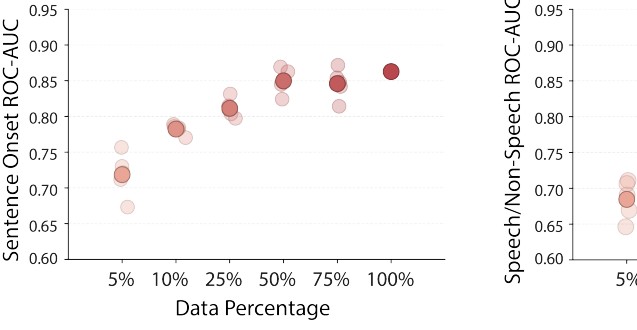 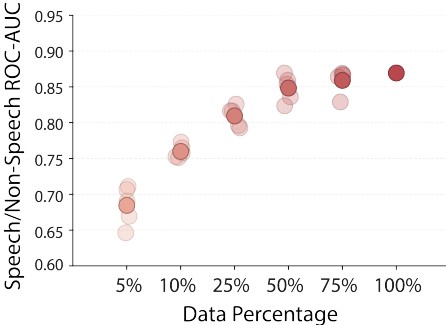

Figure 5: **BaRISTA's downstream classification performance scales as a function of pretraining data size.** Downstream classification results of our best model using different amounts of pretraining data, denoted as a percentage of the full training data (Appendix A). Lighter scatter points represent the average performance of different subsets of training sessions over 5 finetuning seeds; we used 5 different random subsets per percentage. The darker point is the average across these subsets.

## 6 Conclusion

Here we present BaRISTA, a modeling framework that enables flexible use of spatial scales towards spatiotemporal modeling of multiregional intracranial neural activity. First, we introduce a transformer-based model that allows for encoding at larger than channel-level spatial scales. Next, we develop and validate a latent masked reconstruction pretraining task that uses spatial meta-information for masking target tokens, thus also enabling larger spatial scales for masking. We show that utilizing a spatial scale larger than channel-level during pretraining allows our model to improve downstream task performance compared to SOTA iEEG models. Further, the scale of spatial encoding has greater impact on performance than that of spatial masking. Taken together, our results suggest that the choice of spatial scales during masked pretraining, encoding more so than masking, are important for enhanced model performance, especially towards building neurofoundation models of multiregional human intracranial neural activity. Furthermore, by affording flexibility in spatial encoding, our model may serve as a tool to explore hypotheses about the role of brain networks in behavior and cognition.

## Acknowledgments and Disclosure of Funding

This work was partly supported by the National Institutes of Health (NIH) Awards R01MH123770, R61MH135407, DP2-MH126378, and RF1DA056402. We thank Dr. Danil Tyulmankov and Eray Erturk for helpful discussions.

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

## A  Dataset details

Brain Treebank [35] is a publicly available dataset of 10 epilepsy patients collected while they were watching movies from a set of 21 animated/action Hollywood movies. Each subject watched one or more movies while iEEG was being recorded. There is a total of 26 sessions across all subjects, each being 2.07 hours long on average. Electrodes are mapped to common brain atlases that can be used to analyze activity in each brain region. For each session, we first remove corrupted/noisy channels before Laplacian re-referencing the rest, excluding channels with insufficient neighbors for re-referencing – as in [19]. We also additionally removed channels that had been localized to the ventricles of the brain. The final number of channels, parcels, and lobes used per subject is reported in Appendix Table 9. From the 26 available sessions, 17 were used for pretraining, 2 were held out as downstream validation, and the remaining 7 were held out for downstream testing, as specified in Appendix Table 5.

Table 5: List of available sessions in the Brain Treebank dataset, indicating those used for pretraining, downstream validation, and downstream testing. We also report the duration of each session and the number of segments used in pretraining (where relevant).

| Subject | Session | Duration (hrs) | Split | Pretraining Segment # |
|---|---|---|---|---|
| Subject 1 | Session 1 | 1.91 | Pretrain | 1828 |
| | Session 2 | 2.9 | Test | - |
| | Session 3 | 2.07 | Pretrain | 1989 |
| Subject 2 | Session 1 | 2.6 | Pretrain | 2498 |
| | Session 2 | 2.42 | Pretrain | 2342 |
| | Session 3 | 2.66 | Pretrain | 2515 |
| | Session 4 | 3 | Pretrain | 2903 |
| | Session 5 | 3.73 | Pretrain | 3567 |
| | Session 6 | 1.85 | Validation | - |
| | Session 7 | 3.52 | Test | - |
| Subject 3 | Session 1 | 1.9 | Test | - |
| | Session 2 | 2.94 | Pretrain | 2796 |
| | Session 3 | 4.06 | Pretrain | 3924 |
| Subject 4 | Session 1 | 1.87 | Test | - |
| | Session 2 | 1.75 | Pretrain | 1672 |
| | Session 3 | 1.31 | Validation | - |
| Subject 5 | Session 1 | 1.54 | Pretrain | 1482 |
| Subject 6 | Session 1 | 0.81 | Pretrain | 780 |
| | Session 2 | 1.32 | Pretrain | 1267 |
| | Session 3 | 1.6 | Test | - |
| Subject 7 | Session 1 | 1.67 | Test | - |
| | Session 2 | 1.77 | Pretrain | 1696 |
| Subject 8 | Session 1 | 1.41 | Pretrain | 1350 |
| Subject 9 | Session 1 | 1 | Pretrain | 960 |
| Subject 10 | Session 1 | 1.57 | Test | - |
| | Session 2 | 2.33 | Pretrain | 2240 |

For pretraining, we segment data into $T = 3$ second non-overlapping intervals (6144 samples at 2048 Hz), resulting in a total of 35,089 pretraining segments – corresponding to 29.2 hours. The number of pretraining segments per session are reported in Appendix Table 5. We use the same pretraining segments for the channel reconstruction task (Section 4.4). As noted in Section 4.1, for our main evaluation and analyses we generate non-overlapping 3-second segments for the language-related downstream tasks and assign labels using the following protocol: positive-labeled segments are center word-aligned and correspond to sentence onsets or speech whereas negative-labeled samples (for both tasks) are 3-second-long intervals that correspond to no speech content in their entirety; we note

that this definition of negative samples is distinct from the definition used by [14, 19], which only considered the speech content of the center 1-second interval for the label assignment. In Appendix Table 6, we report the number of training, validation, and test segments for each downstream task and hold-out session. Training, validation, and test segments are randomly selected from the generated segments to hit the desired 80/10/10 ratio, similar to [14, 19]. Positive and negative labels were balanced for the classification task before split generation.

In all of our analyses, we z-score standardize the 3-second segments. Further, for both pretraining and finetuning, we generate $n = 12$ temporal patches of length $L = 512$ (corresponding to 250 ms) for each 3-second segment. The subsegment length was chosen based on prior work looking at the timescale of language processing [14, 19, 35, 40], but can be treated as a tunable hyperparameter.

Table 6: For each hold-out session, the number of training, validation, and test segments used in the downstream tasks. Note, these counts correspond to the test sessions in Appendix Table 5.

| Subject | Sentence Onset | | | Speech/Non-Speech | | | Channel Reconstruction | | |
|---|---|---|---|---|---|---|---|---|---|
| | Train | Valid | Test | Train | Valid | Test | Train | Valid | Test |
| Subject 1 | 1488 | 186 | 186 | 2156 | 269 | 269 | 2787 | 348 | 348 |
| Subject 2 | 1036 | 129 | 129 | 1470 | 183 | 183 | 3385 | 422 | 422 |
| Subject 3 | 1066 | 133 | 133 | 1072 | 133 | 133 | 1825 | 228 | 228 |
| Subject 4 | 1022 | 127 | 127 | 1468 | 183 | 183 | 1795 | 224 | 224 |
| Subject 6 | 480 | 60 | 60 | 638 | 79 | 79 | 1532 | 191 | 191 |
| Subject 7 | 650 | 81 | 81 | 624 | 78 | 78 | 1604 | 200 | 200 |
| Subject 10 | 944 | 117 | 117 | 916 | 114 | 114 | 1506 | 188 | 188 |

## B  Model architecture

The temporal encoder in our tokenizer $\mathcal{F}$ was a dilated convolutional neural network (CNN) [29–32] composed of 5 convolutional block layers, with the inner 4 hidden layers having a hidden dimension of 5. Each convolutional block had a kernel width of 3, a stride length of 1, and exponentially increasing dilations as a function of layer depth (i.e., dilation of $2^i$ where $i$ corresponds to the depth, starting from $i = 0$). The CNN operated on univariate channel recordings, and thus the input and final output dimensions were of size 1. Layer normalization was applied on the CNN outputs within each block. A linear layer was then used to transform the CNN output, which is of length $L$, into the final neural tokens with dimensionality $d = 64$.

For our model backbone $\mathcal{G}$, we used an encoder transformer with 12 hidden layers, 4 self-attention heads, and a hidden dimension of $d$ - the same dimensionality as the neural tokens. In each layer, we first apply Root Mean Square (RMS) normalization [41], then perform self-attention followed by a 10% dropout layer, another RMS normalization, and finally a feed-forward MLP. Our predictor network $\mathcal{H}$, used in both the pretraining and downstream channel reconstruction task, is a 5-layer fully-connected network, with 3 hidden layers each followed by a GeLU activation function [42] and a 10% dropout layer. We also use a GeLU activation function after the final layer.

We use a target masking percentage of 30% during pretraining. EMA updates to the target tokenizer $\tilde{\mathcal{F}}$ happened according to a linear warm-up schedule of 10 epochs starting from 0 and increasing to a target momentum of 0.996. In Appendix Table 7, we present the model parameter count for BaRISTA and our baselines (PopT and Brant). Note that despite being significantly smaller than the other two SOTA models (20x smaller than PopT and 500x smaller than Brant), BaRISTA was able to achieve significantly better downstream performance when using larger than channel-level spatial scales. Model code is publicly available at: https://github.com/ShanechiLab/BaRISTA.

## C  Training details

Here we present training details, including computational cost, for both our model and our baseline models. BaRISTA models were pretrained using an effective batch size of 128, with a local batch size of 32 parallelized over 4 NVIDIA RTX 6000 Ada or 4 NVIDIA RTX A6000 GPUs. We used a linear warm-up of 5 epochs to the target learning rate [43], followed by an exponential decay rate of

Table 7: Comparison of the parameter count and total training time between different iEEG models.

| Model | Size | Device | Training Time |
|---|---|---|---|
| BaRISTA | 1M | 4 NVIDIA RTX 6000 Ada | < 4 hours |
| PopT (as reported in [19]) | 20M | 1 NVIDIA TITAN RTX | 2 days |
| PopT (our pretraining, see E.1) | 20M | 4 NVIDIA RTX A6000 | 22 hours |
| Brant (as reported in [15]) | 500M | 4 NVIDIA Tesla A100 | 2.8 days |

$\gamma = 0.99$. We used the AdamW [44] optimizer with a target learning rate of 1e-3 and decay rate of 1e-2 for pretraining. All BaRISTA models were pretrained for 70 epochs, which amounts to 19,500 update steps. PopT pretraining involved 500,000 update steps and Brant reported 750,000 update steps [15].

For the downstream tasks, we again used an effective batch size of 128 for BaRISTA. We finetuned our model for 30 epochs, with a 15-epoch early stopping schedule based on validation performance. As with pretraining, we had a 5-epoch linear warm-up to a target learning rate followed by an exponential decay with decay factor $\gamma = 0.99$. Here, we again used the AdamW optimizer with a decay of 1e-2. Our learning rate was 1e-4 for the pretrained model and 1e-3 for the downstream linear layers. We note that during finetuning we only update the learned spatial encodings, $e^{\mathrm{sp}_w}$ (see Appendix D), and the transformer encoder backbone, while keeping the temporal encoder (dilated CNN) frozen. We empirically found that the difference in downstream classification performance was small when finetuning the CNN as well, and therefore opted to keep the model frozen for the sake of computational efficiency. Randomly initialized versions of our models follow the same downstream learning rate schedules as the pretrained ones.

For finetuning our baselines, PopT and Brant, we matched as closely as possible the training configurations reported in [15] and [19]. For both models, we finetuned for 75 epochs for each hold-out session and used AdamW with decay rate 1e-2. Moreover, for both models we used a linear warmup of 50 update steps to a target learning rate, followed by a step decay scheduler with a step size of 20 updates and decay factor $\gamma = 0.95$. For PopT, the learning rate for the pretrained model was 5e-4 and 5e-5 for the linear classification layer. For Brant, the learning rate was 1e-3 for downstream layers and 1e-7 for the pretrained model. Training batch size for PopT was 128, whereas batch size was 64 for Brant.

Because we ran training for a fixed number of epochs, the total number of finetuning update steps was also dependent on the downstream task and subject (i.e., the number of training segments available), in addition to the batch size. For finetuning on the speech vs. non-speech and sentence onset downstream tasks, the average number of updates for BaRISTA across 7 test sessions was 252, for PopT it was 629 update steps, and for Brant it was 1258 update steps. We chose the larger number of update steps for the baseline models to ensure they converged as we wanted to validate our model's performance against their best performance. We trained Brant the longest as its finetuning learning rate was 1e3 times smaller than BaRISTA's and 1e2 times smaller than PopT's; we note that the learning rates used reflect the rates used by the authors in the original works. Finally, we trained all models using mixed floating-point precision for both pretraining and finetuning.

## D  Spatial scale definitions

Appendix Table 8 defines the within-scale categories discussed in Section 3.2. The number of distinct categories used in our model for each subject can be viewed in Appendix Table 9. For spatial scales consisting of multidimensional spatial information (e.g., the three LPI coordinates), we maintain a distinct embedding table of size $|\mathcal{K}|$ for each dimension (e.g., 3 tables, one for each of the LPI coordinates). The final spatial encoding for a channel is equal to the sum of the embedding vectors corresponding to each dimension: $\mathbf{E}_j = \sum_{w=1}^{|\mathrm{sp}|} e^{\mathrm{sp}_w}(j)$, where $|\mathrm{sp}|$ denotes the number of dimensions in the spatial scale being used and $e^{\mathrm{sp}_w}(j)$ denotes the $w$-th dimension's embedding vector for channel $j$. As an example, our final spatial encoding for the $j$-th channel's LPI coordinates can be expanded as $\mathbf{E}_j = e^{\mathrm{sp}_x}(j) + e^{\mathrm{sp}_y}(j) + e^{\mathrm{sp}_z}(j)$, where each embedding vector $e^{\mathrm{sp}_w}(j) \in \mathbb{R}^d$.

Table 8: Description and examples of the spatial scales defined in Section 3.1. Atlas parcels and lobes categories include hemisphere designations. L/R=Left/Right. Sup.=Superior.

| Spatial Scale | Description | Dimension | Example |
|---|---|---|---|
| Channel | LPI coordinate [24] | 3 | $(x, y, z)$ where each element is an integer between 0 to 200 |
| Atlas Parcels | Discrete neuroanatomical subdivisions of the cortex; subcortical regions also included | 1 | L/R Sup. Temporal Sulcus L/R Postcentral Gyrus L/R Hippocampus |
| Lobes | Brain lobe or equivalently large anatomical region; subcortical regions also included | 1 | L/R Frontal Lobe L/R Hippocampus L/R Temporal Lobe |

Table 9: Number of spatial categories per subject.

| Subject | Channels (LPI Coordinates) | Atlas Parcels | Lobes |
|---|---|---|---|
| Subject 1 | 91 | 25 | 6 |
| Subject 2 | 100 | 25 | 8 |
| Subject 3 | 91 | 20 | 4 |
| Subject 4 | 151 | 36 | 9 |
| Subject 5 | 109 | 25 | 6 |
| Subject 6 | 134 | 30 | 6 |
| Subject 7 | 205 | 47 | 9 |
| Subject 8 | 121 | 29 | 7 |
| Subject 9 | 72 | 19 | 5 |
| Subject 10 | 173 | 42 | 8 |

# E   Experimental details

## E.1   Baselines

We note a few key points with respect to our baseline comparisons.

First, we used the pretrained Brant model as provided by the authors[1]. For PopT [19], we used the publicly available codebase[2] to pretrain PopT. To ensure performance reproducibility, we used the scripts made available by the authors to perform pretraining, black-box, with the only difference being the hardware used (see Appendix Table 7). We verified our pretrained PopT's validity by reproducing the downstream classification results reported in [19]. To do so, we used the publicly available codebase[2] to generate the same train/valid/test splits used in [19] and evaluated our pretrained PopT on the sentence onset and speech discrimination tasks, achieving $0.883 \pm 0.008$ AUC and $0.925 \pm 0.010$ AUC (averaged on 5 finetuning seeds), respectively; the original work reported $0.90 \pm 0.01$ AUC and $0.93 \pm 0.02$ AUC for sentence onset and speech/non-speech, respectively [19].

Second, Brant's model architecture expects temporal patches of length 1500 samples (the original work had pretrained the model using 6-second-long patches at a 250Hz sampling rate). However, the data segments used here were of length 6144 (3-second-long at 2048Hz sampling rate). In order to use the same train/valid/test data segments across all three baselines, we chose to downsample each of our 6144-sample segments to 1500 samples (per segment) before providing them to Brant; we empirically found that this approach worked better than subsegmenting the original segment (i.e., breaking the original 6144-sample segment into 4 subsegments of length 1500 each).

---

[1]Brant Codebase: https://github.com/yzz673/Brant
[2]PopT Codebase: https://github.com/czlwang/PopulationTransformer

### E.2 Classification tasks

For all downstream classification tasks, we use a lightweight linear decoder to evaluate the quality of each model's learned embeddings, as is common practice [45, 46]. To do so, we train a logistic regression with the latent embeddings $\mathbf{Z}$ using a binary cross-entropy loss. For both our model and Brant, we apply a linear projection on all latent embeddings in a sequence to get a single "average" embedding before classification. For PopT, we use the `[CLS]` token as in the original paper [19].

### E.3 Reconstruction task

For the reconstruction task, we perform linear regression from predicted masked tokens $\hat{\mathbf{B}}_{ij} \in \mathbb{R}^d$ to patched neural time-series data, $\hat{\mathbf{P}}_{ij} \in \mathbb{R}^L$, where $d = 64$ is our neural token dimension and $L = 512$ is our temporal patch length. We use our pretrained predictor network $\mathcal{H}$ to generate the predicted masked tokens, such that $\hat{\mathbf{B}}_{ij} = \mathcal{H}(\mathcal{G}(\mathbf{M} + \mathbf{E}_j | \mathbf{S}_{\mathrm{masked}}))$. We finetune our model using a mean-squared error loss between true and reconstructed neural temporal patches, such that

$$\mathcal{L}_{target} = \frac{1}{|\mathbf{B}_{\mathrm{target}}|} \sum_{i \in \{1..n\}, j \in SP_{\mathrm{target}}} \|\mathbf{P}_{ij} - \hat{\mathbf{P}}_{ij}\|_2^2 \tag{1}$$

where $\mathbf{B}_{\mathrm{target}}$ is defined as in Section 3.3 and $n$ denotes the total number of reconstructed patches.

However, using only the predicted masked tokens to learn the mapping from neural tokens to the temporal patches is challenging, as this would require the network to model the true relationship between tokens and patches using a "noisy" (i.e., masked) token prediction. Thus, to help facilitate learning of the mapping from neural tokens to their corresponding temporal patches, we also perform reconstruction of the temporal patches that correspond to the observed ("unmasked") tokens, denoted by $\mathbf{B}_{\mathrm{obs}}$. We compute the mean-squared error for the observed token reconstruction as

$$\mathcal{L}_{obs} = \frac{1}{|\mathbf{B}_{\mathrm{obs}}|} \sum_{i \in \{1..n\}, j \notin SP_{\mathrm{target}}} \|\mathbf{P}_{ij} - \hat{\mathbf{P}}_{ij}\|_2^2 \tag{2}$$

and augment the training loss to be a weighted combination of Equations 1 and 2, such that

$$\mathcal{L} = \mathcal{L}_{target} + \alpha \mathcal{L}_{obs}, \tag{3}$$

where $\alpha$ is an adjustable parameter that regulates the influence of observed (i.e., unmasked) tokens during training. We start with a constant value of $\alpha = 1$ for the first 10 epochs and then linearly decrease it to 0 afterwards. Note that the observed tokens are *only* used during finetuning. For evaluation, we mask out temporal patches one channel at a time, and use the linear head to reconstruct the patches directly from *just* the predicted *masked* tokens, $\hat{\mathbf{B}}_{ij}$.

For this reconstruction task, we used a learning rate of 1e-3 for the pretrained model (predictor network $\mathcal{H}$ included) and a learning rate of 1e-2 for the linear layer. Optimizer scheduling was the same as the classification tasks above (Appendix C). We evaluated on 1 seed per hold-out session and finetuned the models for 20 epochs.

## F Encoding and masking spatial scale analysis

In Appendix Table 10, we present classification performance for the same encoding/masking configurations reported in Table 2, but here we also average across 3 pretraining seeds. As before, we can see that classification performance increases when using larger spatial scales, with parcel-level encoding doing the best on average. Also as before, we see spatial encoding having greater impact than spatial masking.

To better dissociate the impact of each factor on downstream classification performance, we visualize the interaction plots between spatial encoding and spatial masking in Appendix Figure 6. In panels 6A and 6C, we can see that larger than channel-level spatial encoding scales boost downstream classification, across all masking strategies. In panels 6B and 6D, the difference between masking strategies becomes more evident, with the choice of strategy having the greatest impact in the configuration with channel-level spatial encoding.

Table 10: Downstream classification performance of various spatial encoding/masking configurations averaged across all 3 pretraining seeds and 5 finetuning seeds (mean AUC +/- s.e.m.).

| | Mask
Encode | Channels | Parcels | Lobes | Random Init. |
|---|---|---|---|---|---|
| Sentence
Onset | **Channels** | $0.735 \pm 0.013$ | $0.681 \pm 0.012$ | $0.665 \pm 0.012$ | $0.688 \pm 0.017$ |
| | **Parcels** | $0.836 \pm 0.010$ | $\mathbf{0.843 \pm 0.009}$ | $\mathbf{0.844 \pm 0.009}$ | $0.683 \pm 0.017$ |
| | **Lobes** | $0.835 \pm 0.010$ | $0.829 \pm 0.010$ | $0.811 \pm 0.011$ | $0.681 \pm 0.017$ |
| Speech | **Channels** | $0.705 \pm 0.012$ | $0.651 \pm 0.009$ | $0.664 \pm 0.010$ | $0.616 \pm 0.019$ |
| | **Parcels** | $\mathbf{0.847 \pm 0.010}$ | $0.843 \pm 0.010$ | $\mathbf{0.848 \pm 0.010}$ | $0.627 \pm 0.018$ |
| | **Lobes** | $0.837 \pm 0.010$ | $0.829 \pm 0.010$ | $0.806 \pm 0.011$ | $0.628 \pm 0.017$ |

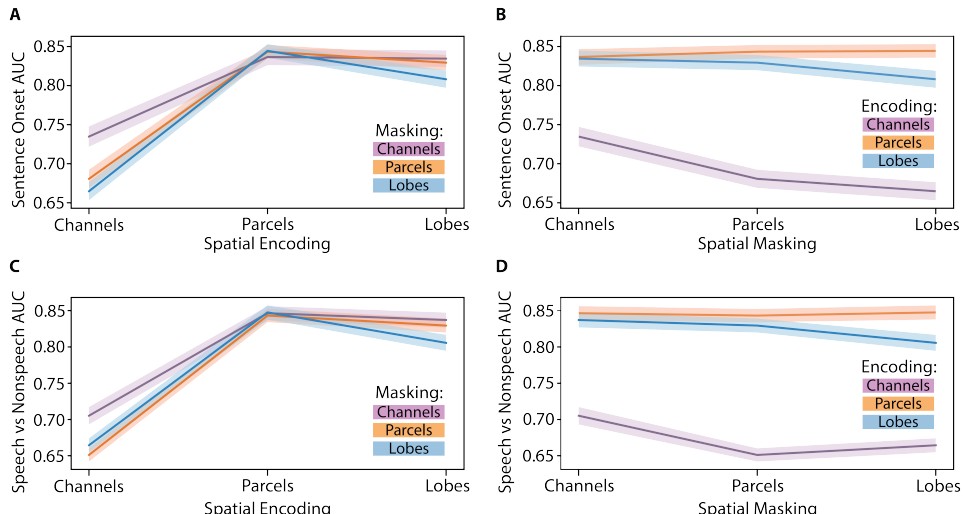

Figure 6: **For all masking strategies, downstream performance on the language tasks improves with greater than channel-level spatial encoding scales, whereas the choice of spatial masking scale has the greatest impact in the configuration with channel-level encoding.** For all panels, solid traces show the average AUC across 3 pretraining seeds and 5 finetuning seeds (shaded areas denote s.e.m.). **A.** Sentence onset classification performance as a function of spatial encoding. Each colored trace corresponds to a different spatial masking strategy, as indicated by the legends. **B.** Sentence onset classification performance as a function of spatial masking strategy; each colored trace corresponds to a different spatial encoding, as indicated by the legends. **C-D.** Same as A-B. but for speech vs. non-speech task.

## G   Interpretability analysis

We also performed an interpretability analysis in which we used the weights of the linear projection that computes an "average" embedding from all latent embeddings during the sentence onset classification task (described in Appendix E.2). By doing so, we aimed to identify the brain regions that our model found to be most critical for decoding sentence onsets. As we detail below, we found that the regions with higher weight loadings indeed corresponded to well-known regions implicated in language tasks, thus suggesting the biological consistency of our learned representations (Appendix Figures 7 and 8).

To perform the interpretability analysis, we first compute the absolute value of the weights, which are of size $nC_{\mathrm{q}}$, where $C_{\mathrm{q}}$ denotes the number of channels in the $q$-th test session and $n$ is the number of temporal patches. Here we had $n = 12$ patches at 250ms for a total duration of 3 seconds (Appendix A). Next, we group channel weights within each of the Destrieux parcels [25] (used here for the sake of visualization) and use the 75th-percentile weight to represent each parcel. Finally, we use session-wise min-max normalization to scale all values to be between 0 and 1. We denote these normalized linear weights by $V^{\mathrm{q}} \in \mathbb{R}^{nR_{\mathrm{q}}}$, with $R_{\mathrm{q}}$ being the number of Destrieux parcels in the test session $q$. We present two different visualizations of these normalized weights (prepared using

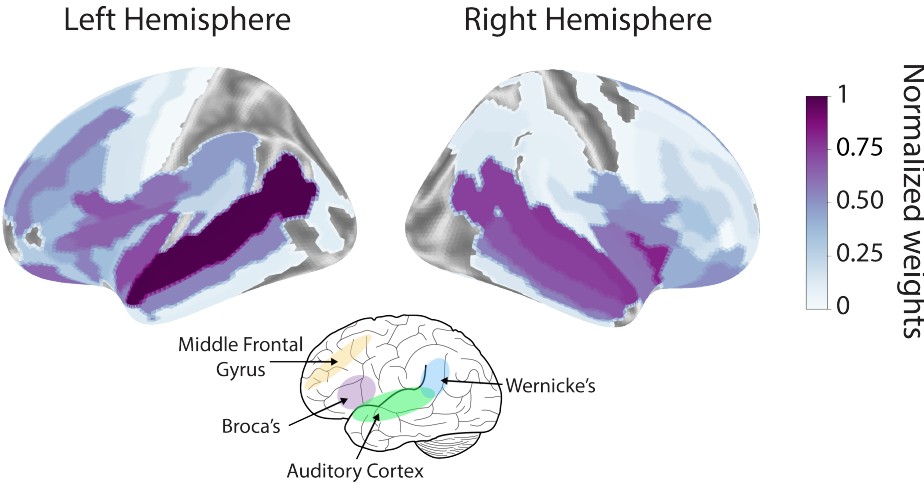

Figure 7: **Normalized linear projection weights have higher loadings on language-related regions across all test sessions**. Weights from our sentence onset classification task are averaged across test sessions and visualized within Destrieux parcels. Bottom visualization depicts the locations of various cortical regions associated with language-related processes.

Nilearn [47]): (1) aggregated across all test sessions and (2) as a function of time (i.e., across the $n$ temporal patches) for a single test session.

For the first visualization, we first aggregate weights across test sessions for each temporal patch, by scaling each session's weights by the associated downstream classification AUC and then forming the weighted average. The aggregated weights, denoted by $V_{\text{agg}}$, allow us to visualize the task-relevant information across the union of all parcels in all test sessions. Lastly, we then average the aggregated weight for each parcel across all temporal patches to compute an average weight per Destrieux parcel, corresponding to each of our 3-second segments. Our results, presented in Figure 7, show larger weight loadings in temporal cortical areas, both in lower-level perceptual regions, such as auditory cortex, as well as in higher-level language processing regions, such as Wernicke's area. Interestingly, we also saw high loadings in the left middle frontal gyrus, which may have language-related implications – as suggested in prior work [48, 49]. These results suggest that our model has learned biologically interpretable embeddings.

In the second visualization, we aimed to better understand the neural dynamics during sentence onset. To do so, in Appendix Figure 8 we visualized the weights for an example test session over time, i.e., over $n = 12$ consecutive 250ms-long temporal patches. We observed an increase in normalized weight loadings for temporal cortical areas shortly after the onset, which corresponds to 0ms in this figure. These results indicate that our embeddings also capture temporal information in the neural data during language tasks.

## H    Subject-specific downstream performance

Per-subject performance for all models on the downstream classification tasks is presented in Appendix Table 11. For BaRISTA we present both the standard pretraining results ("Included" columns) as well as the within-subject generalization results ("Held-out" columns). Subject-specific channel reconstruction results for three of the models reported in Table 3 are provided in Appendix Table 12.

## I    Spectral analysis of channel reconstruction results

After observing the channel reconstruction results presented in Section 4.4, we explored our method's reconstruction in low vs. high frequency ranges. We found that the majority of the spectral power in the reconstructed signal was in the low-frequency range (approximately ≤25Hz on average). For our analysis, we first reconstructed 1162 3-second segments across the 7 test sessions and filtered both the true and reconstructed signals for the low-frequency (<40Hz) and high-frequency (40-

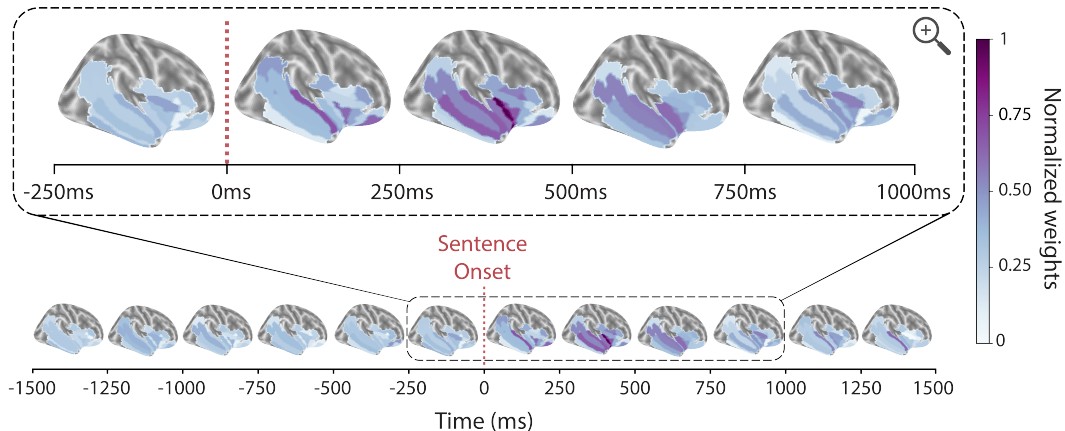

Figure 8: **Normalized weight loadings capture the dynamics of language-processing during sentence onset detection.** We visualize the normalized linear projection weights for a single test session during the course of 3 seconds, where 0ms indicates sentence onset. Weights achieve their maximal values shortly after sentence onset. The inset is a zoomed in version of the dynamics from -250ms to 1000ms relative to sentence onset.

Table 11: Downstream classification results of our model for both standard pretraining and pretraining with the target subject completely omitted (mean +/- s.e.m.).

**Sentence Onset**

| Subject | Included | Held-out | Brant | PopT | Random Init. |
|---|---|---|---|---|---|
| Subject 1 | 0.922 ± 0.002 | 0.921 ± 0.002 | 0.871 ± 0.001 | 0.844 ± 0.007 | 0.818 ± 0.008 |
| Subject 2 | 0.859 ± 0.006 | 0.841 ± 0.011 | 0.771 ± 0.008 | 0.787 ± 0.010 | 0.692 ± 0.006 |
| Subject 3 | 0.956 ± 0.002 | 0.949 ± 0.005 | 0.887 ± 0.009 | 0.898 ± 0.007 | 0.799 ± 0.021 |
| Subject 4 | 0.890 ± 0.006 | 0.841 ± 0.006 | 0.822 ± 0.008 | 0.829 ± 0.007 | 0.619 ± 0.027 |
| Subject 6 | 0.850 ± 0.031 | 0.825 ± 0.024 | 0.635 ± 0.039 | 0.823 ± 0.014 | 0.601 ± 0.036 |
| Subject 7 | 0.686 ± 0.036 | 0.645 ± 0.024 | 0.642 ± 0.009 | 0.622 ± 0.017 | 0.556 ± 0.021 |
| Subject 10 | 0.879 ± 0.008 | 0.868 ± 0.008 | 0.739 ± 0.011 | 0.764 ± 0.009 | 0.695 ± 0.013 |
| **Average** | 0.862 ± 0.015 | 0.841 ± 0.016 | 0.767 ± 0.017 | 0.795 ± 0.014 | 0.683 ± 0.017 |

**Speech/Non-Speech**

| Subject | Included | Held-out | Brant | PopT | Random Init. |
|---|---|---|---|---|---|
| Subject 1 | 0.950 ± 0.003 | 0.942 ± 0.004 | 0.844 ± 0.002 | 0.853 ± 0.006 | 0.808 ± 0.024 |
| Subject 2 | 0.874 ± 0.005 | 0.863 ± 0.004 | 0.745 ± 0.008 | 0.799 ± 0.006 | 0.691 ± 0.013 |
| Subject 3 | 0.977 ± 0.003 | 0.952 ± 0.001 | 0.749 ± 0.012 | 0.897 ± 0.008 | 0.675 ± 0.034 |
| Subject 4 | 0.910 ± 0.004 | 0.863 ± 0.012 | 0.702 ± 0.014 | 0.763 ± 0.010 | 0.505 ± 0.020 |
| Subject 6 | 0.827 ± 0.015 | 0.798 ± 0.011 | 0.624 ± 0.022 | 0.766 ± 0.013 | 0.602 ± 0.027 |
| Subject 7 | 0.708 ± 0.047 | 0.701 ± 0.012 | 0.526 ± 0.029 | 0.591 ± 0.010 | 0.568 ± 0.018 |
| Subject 10 | 0.839 ± 0.01 | 0.843 ± 0.007 | 0.644 ± 0.026 | 0.757 ± 0.020 | 0.540 ± 0.016 |
| **Average** | 0.869 ± 0.016 | 0.852 ± 0.013 | 0.691 ± 0.017 | 0.775 ± 0.016 | 0.627 ± 0.018 |

150Hz) ranges. We then computed the reconstruction error on the filtered signals and compared the performance between the low- and high-frequency ranges. Below we present the results of our analysis for 3 encoding/masking pairs (channels/channels, parcels/parcels, lobes/lobes). We computed the normalized mean-squared error, NMSE (i.e., MSE normalized by the variance of the target signal). We found that this was a better metric for comparing the two regimes since the high-frequency filtered signal had lower amplitude than the low-frequency filtered signal (due to the $1/f$ nature of neural activity). The results are presented in Table 13 and, as expected, the reconstruction error for the high-frequency range is higher than for the low-frequency range.

Table 12: Per-subject channel reconstruction performance for three encoding/masking pairs. Chans=channels.

| Subject | Chans./Chans. | | Parcels/Parcels | | Lobes/Lobes | |
|---|---|---|---|---|---|---|
| | MSE↓ | $R^2$↑ | MSE↓ | $R^2$↑ | MSE↓ | $R^2$↑ |
| Subject 1 | 0.44 | 0.56 | 0.35 | 0.65 | 0.82 | 0.18 |
| Subject 2 | 0.40 | 0.60 | 0.40 | 0.61 | 0.91 | 0.10 |
| Subject 3 | 0.54 | 0.46 | 0.40 | 0.60 | 0.71 | 0.29 |
| Subject 4 | 0.47 | 0.53 | 0.48 | 0.52 | 0.89 | 0.11 |
| Subject 6 | 0.37 | 0.63 | 0.51 | 0.49 | 0.94 | 0.06 |
| Subject 7 | 0.21 | 0.79 | 0.40 | 0.60 | 0.85 | 0.14 |
| Subject 10 | 0.35 | 0.65 | 0.36 | 0.64 | 0.86 | 0.14 |
| mean ± s.e.m. | 0.40 ± 0.04 | 0.60 ± 0.04 | 0.41 ± 0.02 | 0.59 ± 0.02 | 0.85 ± 0.03 | 0.15 ± 0.03 |

Table 13: Channel reconstruction results within low- and high-frequency ranges averaged across 5 finetuning seeds (mean NMSE +/- s.e.m.).

| Model (Encode/Mask) | Low-frequency NMSE↓ | High-frequency NMSE↓ |
|---|---|---|
| Channels/Channels | 0.380 ± 0.005 | 1.370 ± 0.029 |
| Parcels/Parcels | 0.405 ± 0.007 | 1.246 ± 0.015 |
| Lobes/Lobes | 0.879 ± 0.012 | 1.163 ± 0.018 |

# J  Architectural ablations

We performed architectural ablations to evaluate our choice of temporal encoder and interleaved space-time attention. Results are presented below.

## J.1  Choice of temporal encoder

Here we chose to use a dilated CNN as our temporal encoder based on prior works modeling uni/multivariate time-series activity [29–32]. To evaluate this choice, we compared the downstream classification performance of our model when using one of three possible temporal encoders: (1) a dilated CNN (default), (2) a linear projection (i.e., a linear layer the size of our patch length $L$, similar to [15]), and (3) a single layer univariate CNN with kernel size 3 (to match the dilated CNN kernel size). In Appendix Table 14, we present the downstream classification performance on the language tasks (average AUC over 3 pretraining and 5 finetuning seeds) for the parcel/channels encoding/masking configuration presented in Table 1. Our results show that the dilated CNN encoder achieves higher performance than the other two temporal encoders.

## J.2  Choice of combined vs. separate space-time attention modules

We made the decision to use interleaved tokens (i.e., the $\mathbf{S}$ vector) and a single space-time attention module to enable our model to better learn spatiotemporal relationships between channels. Here we empirically show that this interleaved approach outperforms having separated attention modules through an ablation study: we performed an ablation on the $\mathbf{S}$ vector by first passing our sequences through a temporal attention module (i.e., self-attention on the patches within each channel independently) and then passing the output into a spatial attention module (i.e., self-attention on the channels within each patch) – resulting in separated attention modules similar to prior works [15, 18, 19]. For the fairness of comparison, we split our 12-layer transformer into two 6-layer transformers, each with 4 attention heads and the same hidden dimension of 64. In Appendix Table 15, we present the results for the parcels/channels encoding/masking pairing (used in Table 1); AUC scores are averaged across 3 pretraining and 5 finetuning seeds for each of the 7 test sessions. Our results show that the combined attention module achieves higher downstream performance.

Table 14: Downstream classification performance of pretrained parcels/channels BaRISTA models using different temporal encoders (mean +/- s.e.m.).

| Temporal Encoder | Sentence Onset | Speech/Non-Speech |
|---|---|---|
| Linear projection | $0.776 \pm 0.01$ | $0.763 \pm 0.01$ |
| Single layer univariate CNN (kernel=3) | $0.749 \pm 0.013$ | $0.752 \pm 0.012$ |
| Dilated CNN (default) | $\mathbf{0.836 \pm 0.010}$ | $\mathbf{0.847 \pm 0.010}$ |

Table 15: Downstream classification performance of pretrained parcels/channels BaRISTA models using either combined (interleaved) or separate attention modules (mean +/- s.e.m.).

| Attention Module | Sentence Onset | Speech/Non-Speech |
|---|---|---|
| Separate attention | $0.828 \pm 0.010$ | $0.825 \pm 0.011$ |
| Interleaved attention (default) | $\mathbf{0.836 \pm 0.010}$ | $\mathbf{0.847 \pm 0.010}$ |

## K   Extended downstream evaluations on chronological splits and additional tasks

As mentioned in Section 4.1, to extend the evaluation of our model we also performed an alternative (second) evaluation of our main results by generating our downstream segments and creating the train/valid/test splits differently from what was described in Appendices A and E.2 - with the goal of increasing the amount of labeled data available for downstream training.

Our main evaluation used non-overlapping segments that were randomly assigned to train/valid/test splits. Since enforcing no overlap requires dropping some of the annotated segments, in our second evaluation we relaxed the constraint on generating positive-labeled non-overlapping segments for the downstream language tasks, while also generating the train/valid/test splits chronologically in time to avoid any overlap between these splits. Specifically, we again generated 3-second center word-aligned neural segments, but allowed for these segments to overlap. As a reminder, positive here denotes segments that correspond to sentence onset or speech-containing audio; negative-labeled samples were generated as before (Appendix A). By allowing for overlaps, we were able to better utilize the richly-annotated information provided by the Brain Treebank dataset [35] and not restrict ourselves to only a subset of the language-related features. However, to prevent any possible overlap between training and test data due to random split assignments, we instead generated 5 different 80/10/10 train/valid/test splits by partitioning the data chronologically (e.g., the beginning of the recording session for training vs. the end for testing).

In addition to providing a second evaluation for the language-related downstream tasks, this alternative evaluation method provided enough labels for us to also add 2 more downstream tasks: (i) classification of word loudness or softness, and (ii) discrimination of high vs. low magnitude global optical flow in the video stimuli [35]. For these tasks, we again generated center word-aligned segments, each with an associated volume and optical flow measure, and use the top/bottom-quartile approach described in [19] to generate positive and negative labels.

We evaluated the same models from Table 1 on all 4 downstream tasks using the 5 new chronological splits. In Appendix Table 16, we report the average AUC over all test sessions, finetuning seeds, and chronological splits ($n = 175$ points total). The conclusions are the same as before: our model's flexibility in using larger spatial encoding scales during pretraining improved downstream classification performance compared to the baseline models across all tasks. To further verify that our results were consistent with those in Section 4.2, we used a Wilcoxon signed-rank test to assess significance. First, we observed that our channel-level model and PopT were not statistically different in these 4 tasks, but our channel-level model was significantly better than Brant in 3 tasks (p-value$< 1e-3$), i.e., all but the sentence onset task in which they were not statistically different. Second, importantly, our parcels/channels pretrained model was significantly better than both the SOTA baseline models across all 4 tasks (p-value$< 1e-5$) for both Brant and PopT.

We then investigated if the same trends observed in Table 2 regarding the choice of spatial encoding/masking pairs held with the new chronological splits across the 4 downstream tasks. To do so, we evaluated the same 9 models pretrained using distinct spatial encoding/masking combinations

Table 16: Classification results (mean AUC $\pm$ s.e.m.) across 5 chronological split and 5 finetuning seeds. Best-performing model is **bolded** and second-best is underlined model. chans=channels, RI=random initialization.

| Model | Sentence Onset | Speech/Non-Speech | Volume | Optical Flow |
|---|---|---|---|---|
| Brant [15] | $0.772 \pm 0.009$ | $0.650 \pm 0.009$ | $0.571 \pm 0.006$ | $0.531 \pm 0.005$ |
| PopT+Brainbert [19] | $0.776 \pm 0.009$ | $0.724 \pm 0.011$ | $0.584 \pm 0.006$ | $0.551 \pm 0.006$ |
| BaRISTA (chans/chans) | $0.778 \pm 0.009$ | $0.733 \pm 0.011$ | $0.609 \pm 0.007$ | $0.562 \pm 0.006$ |
| BaRISTA (parcels/chans) | **$0.853 \pm 0.007$** | **$0.834 \pm 0.010$** | **$0.698 \pm 0.009$** | **$0.585 \pm 0.006$** |
| BaRISTA (RI, chans) | $0.693 \pm 0.009$ | $0.594 \pm 0.008$ | $0.566 \pm 0.005$ | $0.529 \pm 0.004$ |
| BaRISTA (RI, parcels) | $0.697 \pm 0.009$ | $0.608 \pm 0.009$ | $0.564 \pm 0.005$ | $0.527 \pm 0.003$ |

with the 3 different spatial scales described in Section 3.1. We present both finetuned and random initialization results in Appendix Table 17.

As in our main evaluation, we find that the choice of spatial scale has a significant impact on the performance of the pretrained model, with spatial encoding scale having a greater impact than spatial masking scale. To verify this observation, we again performed a two-way ANOVA [36] with spatial encoding and spatial masking as the independent variables and the AUC values as the dependent variable – Bonferroni correcting p-values to account for the 4 downstream tasks. The results of the ANOVA were consistent with those in the first evaluation, revealing that both independent variables had statistically significant effects on the downstream tasks with only 1 of 4 tasks (optical flow) demonstrating significant interaction between encoding and masking (sentence onset: encoding $p < 1e-3$, masking $p < 1e-3$; speech: encoding $p < 1e-3$, masking $p < 1e-2$; volume: encoding $p < 1e-3$, masking $p < 1e-2$; optical flow: encoding $p < 1e-3$, masking $p < 1e-2$, interaction $p < 1e-2$).

Table 17: Downstream classification results of different spatial encoding/masking configurations (mean AUC +/- s.e.m.) across 5 chronological splits and 5 finetuning seeds. Best results in **bold**.

| Encode / Mask | | Channels | Parcels | Lobes | Random Init. |
|---|---|---|---|---|---|
| Sentence Onset | **Channels** | $0.778 \pm 0.009$ | $0.710 \pm 0.008$ | $0.680 \pm 0.010$ | $0.693 \pm 0.009$ |
| | **Parcels** | **$0.853 \pm 0.007$** | $0.838 \pm 0.009$ | $0.842 \pm 0.008$ | $0.697 \pm 0.009$ |
| | **Lobes** | $0.832 \pm 0.008$ | $0.839 \pm 0.008$ | $0.829 \pm 0.008$ | $0.69 \pm 0.009$ |
| Speech | **Channels** | $0.733 \pm 0.011$ | $0.643 \pm 0.010$ | $0.669 \pm 0.010$ | $0.594 \pm 0.008$ |
| | **Parcels** | **$0.834 \pm 0.010$** | $0.829 \pm 0.010$ | $0.828 \pm 0.011$ | $0.608 \pm 0.009$ |
| | **Lobes** | $0.820 \pm 0.010$ | $0.824 \pm 0.010$ | $0.812 \pm 0.011$ | $0.606 \pm 0.009$ |
| Volume | **Channels** | $0.609 \pm 0.007$ | $0.572 \pm 0.005$ | $0.555 \pm 0.005$ | $0.566 \pm 0.005$ |
| | **Parcels** | **$0.698 \pm 0.009$** | **$0.698 \pm 0.01$** | $0.676 \pm 0.009$ | $0.564 \pm 0.005$ |
| | **Lobes** | $0.693 \pm 0.011$ | $0.683 \pm 0.010$ | $0.677 \pm 0.008$ | $0.565 \pm 0.005$ |
| Optical Flow | **Channels** | $0.562 \pm 0.006$ | $0.527 \pm 0.003$ | $0.519 \pm 0.003$ | $0.529 \pm 0.004$ |
| | **Parcels** | **$0.585 \pm 0.006$** | $0.582 \pm 0.007$ | $0.582 \pm 0.006$ | $0.527 \pm 0.003$ |
| | **Lobes** | $0.581 \pm 0.006$ | $0.578 \pm 0.007$ | $0.571 \pm 0.006$ | $0.529 \pm 0.003$ |

In Appendix Table 18, we report the average number of training, valid, and test samples for each of the 4 tasks when using chronological splits (compare with Appendix Table 6). As before, positive and negative labels were balanced prior to generating the splits.

### K.1 Data scaling and generalizibality of chronological splits

Similar to Section 4.5, we assessed BaRISTA's ability to generalize to completely unseen subjects for our second evaluation method on the sentence onset and speech vs non-speech tasks. Results are provided in Appendix Table 19. Consistent with the first evaluation method (Table 4), we observe a minor performance degradation as expected, while still achieving higher performance compared to baselines. Additionally, we also examined the scalability of downstream performance when pretraining using 5%, 10%, 25%, 50%, and 75% of the total available pretraining data, and observed

Table 18: For each hold-out session, the number of training, validation, and test segments used in the downstream tasks averaged across the 5 chronological splits. Note, these counts correspond to the test sessions in Appendix Table 5.

| Subject | Sentence Onset & Speech/Non-Speech | | | Volume & Optical Flow | | |
|---|---|---|---|---|---|---|
| | Train | Valid | Test | Train | Valid | Test |
| Subject 1 | 2469 | 308 | 308 | 4086 | 510 | 510 |
| Subject 2 | 1626 | 202 | 203 | 2560 | 318 | 319 |
| Subject 3 | 1066 | 132 | 133 | 4048 | 506 | 506 |
| Subject 4 | 1276 | 158 | 159 | 2540 | 316 | 317 |
| Subject 6 | 823 | 102 | 102 | 1000 | 124 | 125 |
| Subject 7 | 650 | 80 | 81 | 4049 | 506 | 506 |
| Subject 10 | 944 | 116 | 117 | 3336 | 416 | 417 |

performance improvement with more pretraining data (Appendix Figure 9) similar to our results for the first evaluation method (Figure 5).

Table 19: Generalizability to new subjects holds for chronological folds: downstream results of our parcels/channels model on chronological folds evaluation for both standard pretraining and pretraining with the target subject completely held-out (mean +/- s.e.m.). Results are averaged across 5 finetuning seeds and 5 chronological folds.

| Model | Sentence Onset | Speech/Non-Speech |
|---|---|---|
| BaRISTA (parcels/channels, Held-out) | $0.841 \pm 0.007$ | $0.819 \pm 0.010$ |
| BaRISTA (parcels/channels, Included) | $0.853 \pm 0.007$ | $0.834 \pm 0.010$ |

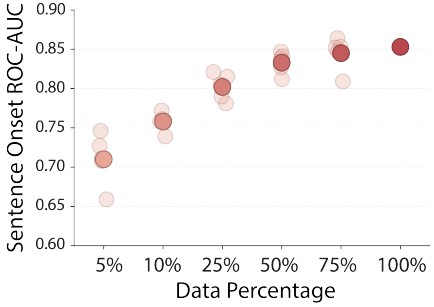 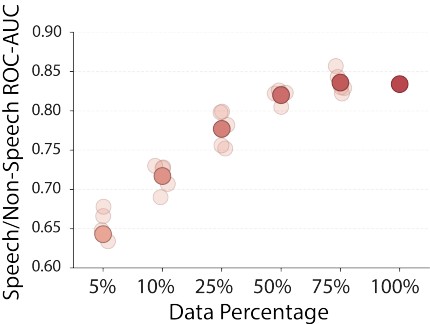

Figure 9: **BaRISTA's downstream classification performance on chronological folds also scales as a function of pretraining data size.** Downstream classification results of our best model using different amounts of pretraining data, denoted as a percentage of the full training data. Lighter scatter points represent the average performance of different subsets of training sessions over 5 chronological splits and 5 finetuning seeds; we used 5 different random subsets per percentage. The darker point is the average across these subsets.

## L   Single-session vs. multi-session models

There has been significant progress on developing models of invasive neural recordings for various modalities such as spikes, local field potentials, and iEEG, for example using state-space models [50–55] or deep learning approaches [56–66]. Many of these approaches have primarily focused on training models for each individual recording session separately. Recently, developing transformer-based neurofoundation models for multi-session training has gotten significant attention for such neural modalities [7–10, 14–19] due to their potential to enable accurate and generalizable modeling of neural datasets by aggregating data across sessions and subjects. Here we show that the scale of spatial encoding and masking are important toward developing neurofoundation models of multiregional human intracranial neural activity and enhancing their downstream decoding performance.

