# OpenReview forum: "BaRISTA: Brain Scale Informed Spatiotemporal Representation of Human Intracranial Neural Activity"
_NeurIPS.cc/2025/Conference — NeurIPS 2025 poster_

### Official Review · Reviewer_eXGi · 2025-06-15

**Clarity:** 2
**Significance:** 2
**Originality:** 2
**Rating:** 3
**Confidence:** 4

**Summary:**

The authors introduce an iEEG-based JEPA-style foundation model, BaRISTA, which explores the impact of different spatial encoding/masking configurations on decoding performance. BaRISTA achieves SOTA performance on Brain Treebank dataset [1].

**Dataset**: The authors evaluate their model on Brain Treebank dataset [1], which includes `10` subjects and `~55` hours iEEG recordings in total. The authors evalutes two sub-tasks, i.e., sentence onset detection and speech/non-speech classification.

**Model**: The authors introduce an iEEG-based JEPA-style foundation model, BaRISTA, whose architecture is similar to LaBraM [2], composing a CNN-based tokenizer and a 12-layer Transformer to model temporal-spatial relationships among iEEG patches.

**Experiment**: BaRISTA explores the impact of different spatial encoding/masking configurations on decoding performance.

**References**:

[1] Chau G, Wang C, Talukder S, et al. Population Transformer: Learning population-level representations of neural activity[J]. ArXiv, 2025: arXiv: 2406.03044 v4.

[2] Jiang W B, Zhao L M, Lu B L. Large brain model for learning generic representations with tremendous EEG data in BCI[J]. arXiv preprint arXiv:2405.18765, 2024.

**Questions:**

See Weaknesses.

**Ethical Concerns:**

["NO or VERY MINOR ethics concerns only"]

**Final Justification:**

I acknowledge the interesting finding of this work, “explor[ing] the impact of different spatial encoding/masking”. As my concerns related to the PopT baseline have not been resolved, I will maintain my score as `3`.

**Limitations:**

yes

**Quality:**

2

**Strengths And Weaknesses:**

**Strengths**:

1. BaRISTA explores the impact of different spatial encoding/masking configurations on decoding performance, and demonstrates that spatial encoding at larger scales than channel-level encoding improves downstream decoding performance.

2. The text has a good structure and is well-written.

**Weaknesses**:

**Major**:

1. The reproduced results for PopT+BrainBERT underperform the original paper [1] -- specifically, `0.80` versus `0.90` on sentence onset detection, and `0.78` versus `0.93` on speech/non-speech classification.

2. I note that the supplementary [code](https://openreview.net/notes/edits/attachment?id=nbRHQko3FE&name=supplementary_material) for PopT [1] on OpenReview implements a lobe-based position embedding approach, which is not described in the PopT manuscript. Given the concerns raised in Major Weakness 1, the contribution of this embedding technique to model performance remains unclear.

**Minor**:

1. A minor implementation difference exists in trial length -- `3` seconds in this work versus the original paper's `5`-second segments.

**References**:

[1] Chau G, Wang C, Talukder S, et al. Population Transformer: Learning population-level representations of neural activity[J]. ArXiv, 2025: arXiv: 2406.03044 v4.

---

> ### Author Rebuttal · Authors · 2025-07-31
>
> We thank the reviewer for their feedback and we reply to the comments inline below.
>
> > The reproduced results for PopT+BrainBERT underperform the original paper [1] -- specifically, 0.80 versus 0.90 on sentence onset detection, and 0.78 versus 0.93 on speech/non-speech classification.
>
> Yes, this is a correct observation. First, we clarify that the reason for this is that we did not use the same data segments used by the authors of Brainbert/PopT [1,2] for our analysis. As such, the reproduction results would not necessarily exactly match. Specifically, those authors generated 5-second overlapping segments which they then partitioned into training/valid/test splits. Here we generated non-overlapping segments to minimize any possible information leakage between segments for a rigorous evaluation of spatial scales.
>
> However, to address the reviewer’s concerns we now demonstrate that our pretrained PopT model can achieve the same performance reported in the original work using our evaluation framework. To do so, we used the publicly available code on GitHub to generate the downstream data splits used in [2] and evaluated our pretrained PopT model on these data splits. In this setup, we achieved **0.883 &plusmn; 0.008 on sentence onset** and **0.925 &plusmn; 0.01 on speech**, which are almost the same as the numbers reported in the original manuscript. We note that the small discrepancies can be due to differences in the evaluation framework, such as seeding and the channels used. In [2] the authors only used 1 finetuning seed (to the best of our understanding) whereas we use 5 to achieve a more robust estimate of model performance. Further, here we use all channels for each session rather than selecting a subset, which is done in [2] (the authors used 90 of the channels).
>
> The above analysis shows that our pretrained PopT model is valid and we are able to use it to reproduce the original reported results.
>
> > A minor implementation difference exists in trial length -- 3 seconds in this work versus the original paper's 5-second segments.
>
> Although this is a valid observation, we clarify that the 3-second vs 5-second distinction is not a limitation of our approach. We chose to use the 3-second interval for two reasons. First, as we noted in the previous response, we chose to generate non-overlapping data segments (distinct from the original work [1,2]). As a result, we would have had less (pre)training data had we used the same 5-second intervals as the original work. Thus we opted to use 3-second intervals. Second, we based our choice of 3-seconds on intervals used in prior works considering language processing [3-5]. For example, the technical paper associated with the Brain Treebank dataset [3] used intervals of 4 seconds for their analysis.
>
> > I note that the supplementary code for PopT [1] on OpenReview implements a lobe-based position embedding approach, which is not described in the PopT manuscript. Given the concerns raised in Major Weakness 1, the contribution of this embedding technique to model performance remains unclear.
>
> We thank the reviewer for directing us to the supplementary code on OpenReview. First, we clarify that we have addressed weakness 1, as we have now shown reproduction of the original numbers (described above). Second, we have integrated the OpenReview implementation of region-based positional embedding into a local copy of the publicly available PopT codebase (from GitHub), and have pretrained a new PopT with region-based encoding. Using the default pretraining settings, the resulting pretrained model underperformed the channel-level encoding version of PopT that was reported in the published paper. We hypothesize that there can be several reasons for this:
>
> 1. The pretraining configurations provided by the default codebase may not be tuned for using parcel-based spatial encoding and thus may require more extensive modification and experimentation, which are beyond the scope of this work. Indeed, the original PopT paper does not report any results with parcel-based encoding, suggesting that they may have seen no improvement with the larger spatial scale.
>
> 2. It could be that the specific discriminative pretraining task used to train PopT benefits from channel-level information more than parcel-level information. For example, the model may have a harder time performing the channel-wise discrimination task if it has the parcel-level encoding rather than a coordinate-level encoding for the channel – as the individual channel’s identity may be critical for the task.
>
> We will revise our manuscript to clarify that our results show the benefit of flexible spatial scales in masked reconstruction pretraining. The impact of spatial encoding in other pretraining approaches (e.g., discriminative tasks such as PopT's) remains an open question and is an interesting direction for future research. Overall, our model enables flexibility in choosing spatial scales for neurofoundation modeling with masked pretraining, shows the importance of considering such scales for strong model performance, and provides the capability for future work to study spatial scales in various downstream motor, cognitive, and sensory tasks.
>
> ** References:**
>
> [1] Christopher Wang et al. BrainBERT: Self-supervised representation learning for intracranial recordings
>
> [2] Geeling Chau et al. Population Transformer: Learning Population-level Representations of Neural Activity.
>
> [3] Christopher Wang et al. Brain Treebank: Large-scale intracranial recordings from naturalistic language stimuli.
>
> [4] Ariel Goldstein et al. A unified acoustic-to-speech-to-language embedding space captures the neural basis of natural language processing in everyday conversations.
>
> [5] Ariel Goldstein et al. Shared computational principles for language processing in humans and deep language models.

---

> > ### Comment · Reviewer_eXGi · 2025-08-01
> > **Official Comment by Reviewer eXGi**
> >
> > Thank your for your efforts to address the points I raised during the rebuttal.
> >
> > > Here we generated non-overlapping segments to minimize any possible information leakage between segments for a rigorous evaluation of spatial scales.
> >
> > Thank you for the valuable feedback. **My primary concern focuses on limited evaluation (e.g., limited supervision samples).** Did you analyze the statistics of word-onsets? I have analyzed the Brain Treebank before. The word interval is extremely short. For example, for subject 01 & session 02, the total word count is `~10k`, and the 5-quantile and 95-quantile of word intervals are **0.07s** and **1.92s**, respectively. Other sessions remain similar. The strict 3-second non-overlapping requirement (Appendix B, Table 2) appears to reduce positive samples significantly (e.g., from `10k` to `<500`), which may underutilize scarce clinical data. **This information is quite important, the authors should place this table in the main text at the initial submission.** I thought the number of samples was similar to that used in PopT’s evaluation. A better evaluation would be adopting chronological K-fold validation (e.g., 80%/10%/10% train/val/test splits by recording time). Alternatively, the division can be finer. Under this splitting strategy, I believe that 5-second samples will not greatly reduce the available samples. Since there is a clear ERP of sentence onset as reported in Brain Treebank [1] Figure 3 (a), I believe K-fold evaluation can maximize the usage of labeled samples, thus providing a fairer comparison with PopT. Additionally, while not required, providing code and checkpoints for reproduction at the initial submission could significantly enhance reproducibility.
> >
> > **References**:
> >
> > [1] Wang C, Yaari A, Singh A, et al. Brain treebank: Large-scale intracranial recordings from naturalistic language stimuli[J]. Advances in Neural Information Processing Systems, 2024, 37: 96505-96540.

---

> > > ### Author Response · Authors · 2025-08-03
> > >
> > > We thank the reviewer for their quick follow-up and careful examination of our responses.
> > >
> > > > limited evaluation (e.g., limited supervision samples)...K-fold evaluation can maximize the usage of labeled samples, thus providing a fairer comparison with PopT
> > >
> > > Based on the reviewer’s great suggestion, below we present K-fold evaluation results which show the same trends as our results in Tab 1 of the manuscript; we will also add these analyses to the manuscript. However, we would first like to clarify why the analysis presented in Tab 1 also allows for a fair comparison between our model and the baseline models:
> > >
> > > 1. All 3 models (Brant, PopT, and BaRISTA) are pretrained using self-supervision without access to any downstream supervision samples or even downstream test sessions. As long as the same evaluation procedure is used across all 3 models, the quality of their learned representations can be fairly assessed and compared agnostic to the total number of downstream supervision samples. Here we use exactly the same downstream data segments for finetuning/evaluating all 3 models, so the total number of supervision samples is exactly the same across all models. As such no model would have an advantage in our evaluation scheme.
> > >
> > > 2. The reviewer noted that, with respect to subject 01+session 02’s data, “the strict 3-second non-overlapping requirement (Appendix B, Table 2) appears to reduce positive samples significantly (e.g., from 10k to <500), which may underutilize scarce clinical data.” We verified that the gap is much narrower than that: the number of positive training samples generated for the speech discrimination task using PopT’s codebase for the subject/session referenced is \~5000 samples vs the \~1000 we used in Tab 1. The gap is even narrower for the sentence onset task: \~1200 positive samples in PopT vs our \~750 positive samples. This shows that we are not in the limited sample/data regime. Indeed a limited data evaluation regime would result in getting close to chance (~0.5 AUC) performance across all the models, which is not the case for sentence onset and speech/non-speech (Tab 1).
> > >
> > > 3. There is significant merit in demonstrating a model’s performance and robustness in scenarios where there is non-dense/scarce clinical data as this is largely the case with intracranial neural recordings and realistic clinical applications wherein supervision samples are not generated from movies but rather from actual patient symptom reporting or evaluation (e.g., tremor, seizure, mood ratings, etc.).
> > >
> > > 4. Our channel-level model performs similarly to PopT, which is another indication of fairness in our evaluation. It is our model with greater than channel-level spatial scales that improves performance over the baseline models and its own channel-level variant, showing the benefit of spatial scales.
> > >
> > > 5. Finally, we would like to note that the goal of this paper is not to show a model that outperforms baselines such as PopT, but rather to develop a model that enables flexibility in encoding/masking at different spatial scales, which allows users to explore the fundamental role of spatial scales in model performance and therefore inform development of neurofoundation models. The reason we compared against PopT & Brant was to demonstrate the validity of our framework by showing that our channel-encoded model performs similarly to these existing SOTA channel-level models.
> > >
> > > Nevertheless we agree with the reviewer that trying a chronological K-fold evaluation would expand our evaluation and thank them for this great suggestion. Below we provide the results for a K=2 cross-validation analysis on the sentence onset task for 5-second segments as suggested by the reviewer. Our results hold in this case too, again showing the importance of considering the impact of spatial scales for developing neurofoundation models.
> > >
> > > |Model|Sentence Onset (5sec)|
> > > |-|-|
> > > |Channels/Chans|0.802 &plusmn; 0.016|
> > > |Parcels/Chans|0.869 &plusmn; 0.013|
> > > |PopT|0.810 &plusmn; 0.017|
> > >
> > > > This information is quite important, the authors should place this table in the main text at the initial submission
> > >
> > > We thank the reviewer for their recommendation. This information was indeed referenced in the main text on lines 242-243 and included in the supplementary material (Tabs 1 & 2). Due to the 9-page limitation of the submission we did not have space to put all tables in the main text. As accepted papers have a 10-page limit, that will allow moving the tables to the main manuscript.
> > >
> > > > providing code and checkpoints for reproduction at the initial submission could significantly enhance reproducibility
> > >
> > > We thank the reviewer for the suggestion. We have every intention to release code after manuscript acceptance as stated in our NeurIPS checklist submission.
> > >
> > > We thank the reviewer again for the follow-up and helpful discussion/suggestions, which allowed us to expand the evaluation of our model, better clarify our contribution, and strengthen our manuscript.

---

> > > > ### Comment · Reviewer_eXGi · 2025-08-03
> > > > **Official Comment by Reviewer eXGi**
> > > >
> > > > Thank you for your efforts to address the points I raised during the rebuttal.
> > > >
> > > > > As such no model would have an advantage in our evaluation scheme.
> > > >
> > > > Regarding sample utilization, we note that Supplementary Table 2 indicates usage of `1190` and `1484` samples on average for Sentence Onset and Speech/Non-Speech fine-tuning, respectively. **This notably exceeds the `500` samples used in PopT [1] (Figure 4), where PopT achieved 0.85 and 0.84 performance on these tasks.** I am curious about the factors contributing to PopT’s significantly lower performance in Table 1.
> > > >
> > > > Besides, collecting a large sEEG dataset from clinical settings is extremely hard, which requires the researchers to maximize data utility. Therefore, **the evaluation settings must be practical in the real world, instead of underutilizing labeled samples**. As reported in BrainBERT [2], BrainBERT utilizes `2844` samples and `7242` samples for downstream fine-tuning on Sentence Onset and Speech/Non-Speech, respectively. Since PopT [1] only modifies the preprocessing pipeline of Pitch & Volume tasks, the total number of samples remains the same in PopT. Based on my experience with the PopT team’s works [1,2,3,4], the preprocessing pipeline remains similar across different works. That’s why I recommend the authors to evaluate chronological K-fold to achieve a fairer comparison with PopT (e.g., 80%/10%/10% train/val/test splits (10-fold) by recording time, instead of your reported 2-fold, which only leverages half of the samples for training). I have run a chronological K-fold evaluation before, and the performance difference was not significant from the original PopT.
> > > >
> > > > > supervision samples are not generated from movies but rather from actual patient symptom reporting or evaluation (e.g., tremor, seizure, mood ratings, etc.)
> > > >
> > > > Thank you for the clarification. **To further strengthen the validity of these claims, I believe experimental validation would be highly beneficial.** Several recent foundation models [1,7,8] have demonstrated performance through comprehensive downstream task evaluations.
> > > >
> > > > > develop a model that enables flexibility in encoding/masking at different spatial scales
> > > >
> > > > Thank you for the clarification. As we all know, this work only evaluates on `2` similar cognitive tasks (i.e., Sentence Onset and Speech/Non-Speech), achieving relatively lower performance and excluding the other tasks in Brain Treebank [2]. **Could the authors comment on how they anticipate the method would outperform other methods and provide meaningful insights in other tasks (e.g., seizure detection [5], and speech decoding [6])?**
> > > >
> > > > **References**:
> > > >
> > > > [1] Chau G, Wang C, Talukder S, et al. Population transformer: Learning population-level representations of neural activity[J]. ArXiv, 2025: arXiv: 2406.03044 v4.
> > > >
> > > > [2] Wang C, Subramaniam V, Yaari A U, et al. BrainBERT: Self-supervised representation learning for intracranial recordings[J]. arXiv preprint arXiv:2302.14367, 2023.
> > > >
> > > > [3] Wang C, Yaari A, Singh A, et al. Brain treebank: Large-scale intracranial recordings from naturalistic language stimuli[J]. Advances in Neural Information Processing Systems, 2024, 37: 96505-96540.
> > > >
> > > > [4] Subramaniam V, Conwell C, Wang C, et al. Revealing vision-language integration in the brain with multimodal networks[J]. ArXiv, 2024: arXiv: 2406.14481 v1.
> > > >
> > > > [5] Zhang D, Yuan Z, Yang Y, et al. Brant: Foundation model for intracranial neural signal[J]. Advances in Neural Information Processing Systems, 2023, 36: 26304-26321.
> > > >
> > > > [6] Zheng H, Wang H, Jiang W, et al. Du-IN: Discrete units-guided mask modeling for decoding speech from Intracranial Neural signals[J]. Advances in Neural Information Processing Systems, 2024, 37: 79996-80033.
> > > >
> > > > [7] Wang J, Zhao S, Luo Z, et al. Cbramod: A criss-cross brain foundation model for eeg decoding[J]. arXiv preprint arXiv:2412.07236, 2024.
> > > >
> > > > [8] Yuan Z, Shen F, Li M, et al. Brainwave: A brain signal foundation model for clinical applications[J]. arXiv preprint arXiv:2402.10251, 2024.

---

> > > > > ### Author Response · Authors · 2025-08-05
> > > > >
> > > > > We thank the reviewer for their prompt follow-up. We understand that the reviewer’s primary concern is analyses with one of the baselines, namely PopT. Before addressing these concerns, we believe there may be a miscommunication about our claims that is important to clarify. We reiterate that the contribution of our work is _not_ a neurofoundation model that aims to outperform baselines. **Rather our goal is to design a model and pretraining framework that can enable the flexible assessment of spatial encoding/masking scales within the context of a masked reconstruction pretraining task.** Based on our experimental findings, our primary claim is that flexibility in spatial scales is important for neurofoundation models pretrained with a masked reconstruction task. While the language-related tasks here show that greater than channel-level spatial scales perform more strongly, there may very well be scenarios wherein channel-level encoded models (such as our baselines and our channel/channel model) achieve better performance (e.g., for channel reconstruction, sec 4.3). As such, our claim is that incorporating flexibility in spatial scales is an important consideration toward designing neurofoundation models. We will further clarify these points in the manuscript.
> > > > >
> > > > > Now we provide evidence for why the PopT results are obtained appropriately and are not at odds with the original manuscript:
> > > > >
> > > > > 1. **Number of supervision samples is sufficient.** We appreciate the reviewer now noting that the number of samples we used for the two downstream tasks “notably exceeds the 500 samples used in PopT [1] (Fig 4)”. We are glad the reviewer agrees that the number of samples used in Tab 1 exceeds that required by PopT to achieve close to full performance, indicating that our analyses are not in the limited data regime.
> > > > >
> > > > > 2. **We did follow an 80/10/10 split.** For the results presented in our last response, we indeed followed the reviewer’s recommendation and used **2 different 80/10/10 train/val/test chronological splits** to evaluate the models, thus using **80% of the data for training not half**. We have since added 3 more (different) chronological 80/10/10 train/val/test splits for sentence onset with the following average results across the 5 splits:
> > > > >
> > > > > |Model|Sentence Onset|
> > > > > |-|-|
> > > > > |PopT|0.805 &plusmn; 0.001|
> > > > > |Channels/Chans|0.787 &plusmn; 0.01|
> > > > > |Parcels/Chans|0.858 &plusmn; 0.008|
> > > > >
> > > > > Our previous observations again hold in these results: (1) our channel-encoding model performs comparable to PopT (another channel-level model) while (2) the use of larger spatial encoding scales during pretraining helps with downstream performance.
> > > > >
> > > > > 3. **We reproduced the PopT manuscript numbers on the original data segments.** We showed in our first rebuttal response that on the original PopT segments our pretrained PopT model gets numbers similar to [1], again showing validity.
> > > > >
> > > > > 4. **Our channel/channel BaRISTA model performs similarly to PopT, a channel-level encoding model.** This is indeed how we verified the validity of our pretraining framework and model architecture: we wanted to confirm that we can achieve comparable channel-level performance against strong SOTA models such as PopT & Brant.
> > > > >
> > > > > > curious about the factors contributing to PopT’s significantly lower performance in Tab 1
> > > > >
> > > > > As noted in item 3 above, we were able to reproduce AUC numbers similar to the original manuscript when using the original data segments/splits. As such, the difference in numbers reported in Tab 1 is likely due to how we generated the non-overlapping (vs overlapping) neural segments and to differences in the evaluation procedure (number of channels and finetuning seeds) that were described previously.
> > > > >
> > > > > > downstream task evaluations
> > > > >
> > > > > To further expand our model’s evaluation and show the importance of spatial scales for more tasks, we added decoding of 2 more features (i.e., 2 more tasks) provided in the Brain Treebank dataset (volume and global optical flow magnitude) [2], as per recommendations from reviewers uiXd and 54tZ. We used the same 80/10/10 chronological splitting approach as above and the results for a single split are presented in our rebuttals to reviewers uiXd and 54tZ; we have since run an additional different split for these tasks and gotten similar results, which we will add to the manuscript.
> > > > >
> > > > > In summary here we propose a new model and masked reconstruction pretraining framework that enables flexible assessment of encoding/masking spatial scales. Our empirical results demonstrate that the choice of spatial scales during masked reconstruction pretraining is important and should be evaluated and selected carefully for the targeted use case. We thank the reviewer for their attention to detail and the engaging and helpful discussion.
> > > > >
> > > > > [1] Chau G, et al. Population transformer: Learning population-level representations of neural activity.
> > > > >
> > > > > [2] Wang C, et al. Brain treebank: Large-scale intracranial recordings from naturalistic language stimuli.

---

### Official Review · Reviewer_uiXd · 2025-06-16

**Clarity:** 3
**Significance:** 3
**Originality:** 3
**Rating:** 5
**Confidence:** 4

**Summary:**

In this work, the authors present a self-supervised pretraining framework for stereotactic EEG (sEEG) data. Unlike prior methods, their approach introduces a flexible mechanism for incorporating spatial information about electrode locations, allowing the model to be informed at varying levels of granularity, ranging from individual electrodes to anatomical parcels or entire lobes. Another innovation of the framework is a novel reconstruction-based task in which the reconstruction targets are selected based on spatial meta-information, rather than through random sampling. The authors evaluate their method on the publicly available Brain Treebank dataset. They demonstrate that their pretraining strategy enhances decoding performance when finetuned on two downstream tasks, surpassing the performance of leading pretraining baselines, including PopT and Brant. Further analysis reveals that the spatial scale at which location information is encoded significantly affects downstream performance: encoding at the parcel level yields better results than encoding at the electrode level. Importantly, they also show that parcel-level representations retain nearly all channel-level information, as evidenced by performance on a channel-level reconstruction task. In supplementary experiments, the authors show that their framework scales with more training data and that the learned representations transfer effectively to subjects not seen during pretraining.

**Questions:**

Major:
1. Brain Treebank is a dataset that has labels to enable testing your framework on more downstream tasks than the ones used in this work. For example, PopT also evaluates their model on volume, and pitch discrimination tasks as well. Please justify why you did not evaluate your model on those tasks or evaluate the utility of your pretrained model on those tasks as well. See details here: https://arxiv.org/abs/2406.03044.
2. Based on Table 3 of the Supplement, it seems like your model is substantially smaller than the baselines it is compared against (Brant & PopT). Please justify why you did not parameter match your model with your baselines or even better, report the results shown in table 1 of the main text after training versions of Brant and PopT that have approximately the same number of parameters as your model.
3. Based on Figure 4 of the main text, it looks like your pretraining strategy does a good job at capturing the low-frequency information in the sEEG signal. However, it looks like the model is doing a bad job at capturing the high-frequency details of the signal. This is a common problem encountered when pretraining on reconstruction tasks with MSE Losss (since there is a ~1/f relationship between frequency and power in biological signals). Note: This observation might be exacerbated due to the Savitzky-Golay filter used for plotting purposes.
    1. Do you have a sense of whether your modelling approach is better at capturing high-frequency information (i.e. frequency in the range 70-150 Hz) compared to Brandt and PopT? A relatively easy way to check this would be: using the pretrained model, (1) reconstruct a large amount of sEEG traces (similar to what you describe in section 4.4 of your main text), (2) filter the reconstructed and ground truth signal in the frequency range 70-150 Hz (and notch filter line noise, if necessary), (3) calculate the MSE between the reconstructed and ground truth signals. Report the results of this procedure for your model as well as Brant and PopT.
    2. I believe this work would benefit by mentioning this limitation in the discussion as a directions for future work.

Minor
1. Throughout the text, the authors use \mathcal{R} to denote the set of real numbers. I believe a more standard notation would be to use \mathbb{R}. Consider replacing.
2. It would be helpful for the reader if you included in the captions of tables (or at least of table 1) what does bold, underline, and * indicate.
3. Typo in lines 10-11 of Supplement. I believe the main text mentions that 17 sessions were used for pretraining but supplement mentions 16.
4. Please explain why you chose to not show Table 7 of the supplement in place of Table 2 in main. Consider replacing Table 2 in main with Table 7 of supplement, as the results shown in Table 7 of supplement are a more robust estimate of your model's performance.
5. The results shown in Supplement E and F are really strong. I believe that moving them to the main text on your manuscript (possibly after revisions where you get an additional page for the main text) would help the reader appreciate the utility of your approach.

**Ethical Concerns:**

["NO or VERY MINOR ethics concerns only"]

**Final Justification:**

After careful consideration of the authors’ rebuttal, I have decided to increase my score and recommend that this paper be accepted for publication.

From the initial submission, I believed that this work addressed an important and unresolved problem in neural decoding: how to best incorporate spatial information into models. The authors provide compelling evidence that parcel-level information may be more important than channel-level information.

During the rebuttal, the authors addressed two key concerns:
1. They demonstrated that their modeling approach captures low-frequency information better than high-frequency information, a common limitation for neural decoding models, and committed to including this point in their limitations and discussion section.
2. They conducted additional experiments showing that their models perform on par with PopT and Brant.

Overall, I believe this work offers valuable insights into an important, previously unaddressed problem in neural decoding: determining the optimal spatial scale for incorporating information into models. The results presented here can be highly useful for the field, and therefore I recommend this paper be accepted for publication.

**Limitations:**

Yes. For a suggestion please see major question 3.2.

**Quality:**

3

**Strengths And Weaknesses:**

Strengths:
1. Flexibility to incorporate spatial information at different spatial scales. To my knowledge, parcel or lobe level encoding has not been used to inform models with spatial information during pretraining before.
2. Insight into the spatial scale of encoding. The majority of previous works have encoded spatial information at the electrode level. However, the results shown in this work support that encoding information at larger scales (parcel level) result in better models. This could steer the field towards adopting a better positional encoding scheme (tailored to sEEG) in future works.
3. The introduced framework + architecture scales to large amounts of data and is transferable to new subjects.
4. The manuscript is well written and easy to follow.

Weaknesses:
1. Limited evaluation. While the authors show their pretraining strategy on a large dataset, they only evaluate the utility of the learned representation on 2 downstream decoding tasks.
2. Unclear whether this method is good at capturing "high-frequency" information of sEEG singals (see major question 3).

---

> ### Author Rebuttal · Authors · 2025-07-31
>
> We thank the reviewer for their feedback, questions, and comments. We reply inline below.
>
> > Limited evaluation
>
> This is an important point. There are two reasons for not evaluating on those tasks. First, we clarify that the primary focus of our work was to assess the impact of different spatial scales during pretraining on downstream model performance. As such, we prioritized performing comparisons of our pretraining framework under different spatial encoding/masking configurations, rather than demonstrating our model’s utility/generalizability on a variety of downstream tasks. Second, we did not have sufficient labeled data for a per-subject pitch and volume discrimination task to enable robust model training. This is because we generated our downstream data segments differently than how the authors of PopT generated their data segments: as noted in our manuscript, we controlled our data splits to use *non-overlapping* segments to avoid any potential information leakage between training and test splits (especially since neural data is highly correlated over time), therefore enabling a rigorous evaluation of spatial scales. In contrast, the original segments in PopT did contain overlap. Because of our stringent no overlap requirement, combined with the quartile label generation (as used in [1,2] for the pitch/volume tasks), we did not have sufficient training samples left over to reliably train the model. Indeed, we observed chance performance across all models (PopT, Brant, our variants) for pitch/volume discrimination due to the small training set size. That is why we decided against evaluating on pitch/volume and instead focused on the speech/non-speech and sentence onset tasks.
>
> However, to address the reviewer’s concern about limited evaluation we decided to modify our segmentation scheme to generate more labels, while still addressing the information leakage concern. To do so, we modified our scheme to allow overlapping segments but selected train/valid/test splits across time (i.e., the beginning of the recording session for training vs. the end for testing), instead of randomly shuffling the segments and then selecting splits – as is done in PopT’s segment generation. Based on Fig. 9 of the Brain Treebank technical paper [3], we decided to decode volume which showed high SNR across multiple brain regions. We present our results (average AUC across 7 test sessions, 5 seeds each) using PopT and our pretrained models from Tab. 1 (channels/channels and parcels/channels). As we can see, similar trends on the effect of spatial scales hold for this downstream task as well, thus expanding our model’s evaluation.
>
> |Model|Volume|
> |-|-|
> |Channels/Channels|0.656 &plusmn; 0.015|
> |Parcels/Channels|**0.714 &plusmn; 0.016**|
> |PopT|0.639 &plusmn; 0.009|
>
> > "high-frequency" information of sEEG signals; is your model better at capturing high-frequency information compared to Brant and PopT; mention limitation in the discussion
>
> We thank the reviewer for their excellent suggestion of an interesting analysis. We looked into our method’s ability to capture high-frequency information and indeed observed that the majority of the spectral power in the reconstructed signal was in the low-frequency range (approximately <=25Hz on average). As per the reviewer’s recommendation we performed the following analysis: we reconstructed 1162 3-second segments across the 7 test sessions which we low-pass filtered for the low-frequency (<40Hz) range and band-pass filtered for the higher frequency (40-150Hz) range. We then computed the reconstruction error on the filtered signals and compared the performance between the low and high-frequency ranges. Below we present the results of our analysis for 3 encoding/masking pairs (channels/channels, parcels/parcels, lobes/lobes). We computed the normalized MSE (i.e., MSE normalized by the variance of the target signal), which we found was a better metric for comparing the two regimes since the high-frequency filtered signal had lower amplitude than the low-frequency filtered signal (due to the 1/f relationship noted by the reviewer). As the reviewer hypothesized, the reconstruction error for the high-frequency range is higher than for the low-frequency range.
>
> |Model|Low-frequency NMSE $\downarrow$|High-frequency NMSE $\downarrow$|
> |-|-|-|
> |Channels/Channels|0.380 &plusmn; 0.005|1.370 &plusmn; 0.029|
> |Parcels/Parcels|0.405 &plusmn; 0.007|1.246 &plusmn; 0.015|
> |Lobes/Lobes|0.879 &plusmn; 0.012|1.163 &plusmn; 0.018|
>
> We also compared against Brant by finetuning Brant for the channel reconstruction task, but model convergence was very slow and the performance was poor. We did not compare against PopT because we do not expect PopT to have the ability to faithfully reconstruct the channel activity. This is because PopT’s default temporal encoder (pretrained Brainbert) summarizes the input segments by computing an average temporal representation using the middle ~500ms (center 10 samples) – regardless of the input segment length. As such, we don’t believe the model has sufficient temporal information to reconstruct the channel activity for the full 3-second interval. We will revise the discussion section to explicitly make note of the limitation of high-frequency reconstruction and list it as a direction for future work. We will also include a more comprehensive supplementary section with the analysis discussed above.
>
> > justify why you did not parameter match your model with your baselines
>
> This is also a good point. First we clarify that our primary goal in this work, as stated in the first response, was to investigate the impact of spatial scales on model performance. As such, we optimized our model to be lightweight enough to permit multiple pretrainings in a reasonable amount of time (to test the various encoding/masking combinations), while still maintaining sufficient modeling capacity to perform well on downstream tasks. This is why we did not parameter match the larger models. Indeed, the primary reason for including the comparisons against the baseline models in Tab. 1 was to demonstrate our proposed model’s & pretraining framework’s validity (by showing that we are able to achieve results comparable to prior published works when we used channel-level encoding).
>
> The reviewer’s suggestion to train tiny versions of Brant and PopT is interesting. We do not have access to the original dataset the larger Brant model was pretrained on and thus don’t think it would be suitable to train smaller versions of Brant without this dataset. Moreover, Brant's pretraining code is not publicly released for us to pretrain a tiny version on a different dataset. However, having access to PopT's codebase (on GitHub) and the Brain Treebank dataset (used to pretrain PopT), we did try pretraining a smaller version of PopT. We identified a model configuration that allowed for ~1M (1790594) parameters by changing the hidden dimension from the default 512 to 64. However, pretraining using the default pretraining configurations provided by the authors was unsuccessful; we evaluated downstream performance on one task for a single test session and found the performance was close to chance. We also tried a ~3M (3772930) parameter model (hidden dimension 128), but pretraining was again unsuccessful with their default settings. Pretraining a tiny PopT may require more substantial changes to the pretraining configuration and/or the model configuration itself (i.e., number of layers, etc.) than is within the scope of this work. Finally, the larger Brant and PopT models are likely better than their tiny versions on the downstream tasks, as is generally the case with foundation models. Thus, we used the best possible performance from our baselines as a comparison point for our model.
>
> > explain why you chose to not show Table 7 of the supplement in place of Table 2 in main; consider replacing Table 2 with Table 7
>
> We thank the reviewer for their question/comment. The primary reason for showing the best pretraining seed in Tables 1 and 2 was to maintain consistency with common practice and for fairer comparison with our baselines. In Table 1, we chose a single model to fairly compare against our baseline models which reflected just one pretraining seed (i.e., a single model); we used validation set decoding to select the model. Then, for the sake of consistency, we used the same criteria to select the best performing model within each category for Table 2. Finally, for completeness we presented the most robust estimate of our model’s performance for each encoding/masking combination in Table 7. Thus, we ended up choosing the simplified Table 2 to present in the main manuscript largely for ease of exposition/explanation and to maintain consistency with the criteria used for Table 1. Our results and conclusions are consistent in both Tables 2 and 7. We will additionally provide Table 7 in the main text of our manuscript following the reviewer’s suggestion.
>
> > Replace $\mathcal{R}$ with $\mathbb{R}$; include captions of tables (or at least of table 1); typo in lines 10-11 of supplement
>
> We thank the reviewer for their recommendations and for catching the typo. We will make the suggested changes to improve readability of the text and to be more aligned with standard notation.
>
> > Supplement E and F are really strong; move them to the main text
>
> We thank the reviewer for describing our results to be strong and for the suggestion. Upon manuscript acceptance we will move supplements E and F into the main manuscript.
>
> **References**
>
> [1] C. Wang et al. BrainBERT: Self-supervised representation learning for intracranial recordings.
>
> [2] G. Chau et al. Population Transformer: Learning Population-level Representations of Neural Activity.
>
> [3] C. Wang et al. Brain Treebank: Large-scale intracranial recordings from naturalistic language stimuli.

---

> > ### Comment · Reviewer_uiXd · 2025-08-01
> >
> > Thank you for the detailed responses to all my comments/questions. The clarifications your provided, as well as the additional experiments boosted my confidence in the quality and impact of your work, which is why I will be raising my score to a 5/6 and recommending the manuscript be accepted for publication.
> >
> > I still have a few minor concerns/suggestions, which I summarize below.
> >
> > > "This is because we generated our downstream data segments differently than how the authors of PopT generated their data segments: as noted in our manuscript, we controlled our data splits to use non-overlapping segments to avoid any potential information leakage between training and test splits (especially since neural data is highly correlated over time), therefore enabling a rigorous evaluation of spatial scales. In contrast, the original segments in PopT did contain overlap. Because of our stringent no overlap requirement, combined with the quartile label generation (as used in [1,2] for the pitch/volume tasks), we did not have sufficient training samples left over to reliably train the model. Indeed, we observed chance performance across all models (PopT, Brant, our variants) for pitch/volume discrimination due to the small training set size. That is why we decided against evaluating on pitch/volume and instead focused on the speech/non-speech and sentence onset tasks."
> >
> > Thank you for this important clarification. I believe it would be worth mentioning this in the discussion of your paper, as it would show the reader that those results are not missing because your model performed worse than other frameworks, rather, because of limitations inherent to the dataset. Being transparent about this would increase the reader's confidence in your results.
> >
> > > " We present our results (average AUC across 7 test sessions, 5 seeds each) using PopT and our pretrained models from Tab. 1 (channels/channels and parcels/channels). As we can see, similar trends on the effect of spatial scales hold for this downstream task as well, thus expanding our model’s evaluation."
> >
> > Thank you for those additional experiments which increase my confidence in your framework's performance. Could you please clarify why you compared against PopT only as opposed to also comparing against PopT and Brant for this particular experiment?
> >
> > > explain why you chose to not show Table 7 of the supplement in place of Table 2 in main; consider replacing Table 2 with Table 7
> >
> > Thanks for clarifying why you kept the results from only one training seed (Table 2) in the main text. Based on this insight, I agree that keeping Table 2 in the main text and maybe referencing Table 7 is a good choice.
> >
> > Thanks again for your efforts and I am looking forward to receiving your responses.

---

> > > ### Author Response · Authors · 2025-08-03
> > >
> > > We thank the reviewer for their prompt response and careful evaluation of our explanations and additional experimental analyses. We greatly appreciate the score increase and recommendation for acceptance.
> > >
> > > > I believe it would be worth mentioning this in the discussion of your paper, as it would show the reader that those results are not missing because your model performed worse than other frameworks, rather, because of limitations inherent to the dataset. Being transparent about this would increase the reader's confidence in your results.
> > >
> > > We thank the reviewer for their recommendation and will include this clarification in the revised manuscript. We currently explain our segment generation in lines 230-243. In that same section (4.1), we will explicitly discuss why our segment generation scheme compelled us to limit our evaluation to two downstream tasks. As the reviewer stated, adding this information will further clarify the process and raise confidence in results.
> > >
> > > > Could you please clarify why you compared against PopT only as opposed to also comparing against PopT and Brant for this particular experiment?
> > >
> > > We thank the reviewer for the clarifying question. Given the time-constraints for the rebuttal period, we focused on PopT as it has generally been shown to achieve higher performance than Brant both in the PopT paper and in our manuscript. However, we have now evaluated Brant on the same task and gotten 0.554 +- 0.010 AUC, which is lower than both PopT’s and our model’s performance as expected.
> > >
> > > > I agree that keeping Table 2 in the main text and maybe referencing Table 7 is a good choice.
> > >
> > > Thank you for acknowledging our rationale. We will be sure to clearly reference Tab. 7 in the main manuscript.
> > >
> > > Once again, we thank the reviewer for the helpful and very constructive feedback and discussion that we believe will greatly improve the quality of our manuscript and make our results stronger and more compelling. We also appreciate that they will raise their score and recommend acceptance.

---

### Official Review · Reviewer_54tZ · 2025-07-03

**Clarity:** 3
**Significance:** 3
**Originality:** 3
**Rating:** 4
**Confidence:** 4

**Summary:**

The authors develop a model, BaRISTA, in which varying spatial scales are used to decode iEEG activity in a complex task. They achieve noticeable improvements in performance via flexibility in larger spatial encoding scales, a novel addition to previous models using only channel-level spatial encoding. Instead of randomly selecting channels during model training, the authors chose spatial scales for the masked reconstruction in latent space, thus resulting in a high fidelty model.

**Questions:**

Do spatial scales vary given the classification task/experimental stimuli? E.g. smaller spatial scale needed to encode visual features in environments vs larger spatial scale for more cognitively demanding tasks such as the language tasks here?

Although the current model performs quite well, it would be interesting to test if the same performance gains could be achieved using a second dataset in which people engage in a far less cognitively demanding task (such as viewing images of scenes). I.e. Are the benefits of varying spatial encoding less advantageous in situations where shorter, less flexible temporal and spatial scales are needed to detect changes in one's environment?

**Ethical Concerns:**

["NO or VERY MINOR ethics concerns only"]

**Limitations:**

Yes

**Quality:**

3

**Strengths And Weaknesses:**

### Pros:

* One of the major achievements of this model is that the authors consider varying spatial scales of human brain (via iEEG) in modeling neural activity, which is very important when capturing high-level cognitive processes.

* The authors use a dataset well-suited to capture spatially complex brain patterns. Movie watching, even at shorter timescales, give rise to high variability within and across individuals' spatial and temporal patterns of activity.

* Varying spatial encoding substantially outperforms other models and non-variable encoding methods.

### Cons:

* Please clarify earlier on in your intro the working definition of neural tokenization. Although the context throughout the paper resolves some questions about the term, an earlier definition would improve readability.

* The current model was tested on a single dataset, which is understandable given the nature of iEEG datasets. As it stands, I wonder if  the flexibility of spatial encoding is only marginally advantageous in a task that is less cognitively demanding in which rich information processing is not needed. A simulation testing the current model's parameters and the necessity of spatial encoding flexibility could be very useful here.

---

> ### Author Rebuttal · Authors · 2025-07-31
>
> We thank the reviewer for their recommendations, questions, and comments. We address these below inline.
>
> > Do spatial scales vary given the classification task/experimental stimuli? E.g. smaller spatial scale needed to encode visual features in environments vs larger spatial scale for more cognitively demanding tasks such as the language tasks here? …it would be interesting to test if the same performance gains could be achieved using a second dataset in which people engage in a far less cognitively demanding task (such as viewing images of scenes). I.e. Are the benefits of varying spatial encoding less advantageous in situations where shorter, less flexible temporal and spatial scales are needed to detect changes in one's environment?
>
> This is a great question and will be an interesting hypothesis to test in follow-on work. Based on the analyses performed here (i.e., the language-related decoding task and the channel reconstruction task), we hypothesize that the reviewer’s expectation may indeed be the case and that different applications will benefit from different spatial encodings – which can be assessed by incorporating and testing each scale using our model and its flexible spatial encoding capability. For example, we observed that models pretrained using smaller encoding spatial scales were able to perform channel-level neural reconstruction more accurately than models pretrained with larger encoding spatial scales. On the other hand, larger spatial scales helped more with language-related classification tasks.
>
> To address the reviewer’s question, we decided to try decoding one of the visual features (global optical flow magnitude) provided by the Brain Treebank dataset [1]. We used the two encoding/masking configurations highlighted in Table 1 (i.e., parcel/channels and channels/channels), and found that the parcel/channels model outperformed the channel/channel model. Our results are presented in the table below (average AUC across the 7 test sessions and 5 finetuning seeds each); as baselines we compared against PopT and randomly initialized BaRISTA models with channel-level and parcel-level encoding.
>
> |Model|AUC &plusmn; standard error of measure|
> |-|-|
> |Random initialization (channels)|0.525 &plusmn; 0.008|
> |Random initialization (parcels)|0.527 &plusmn; 0.006|
> |Channels/Channels|0.561 &plusmn; 0.010|
> |Parcels/Channels|**0.604 &plusmn; 0.013**|
> |PopT|0.549 &plusmn; 0.009|
>
> These results indicate that the flexibility of spatial encoding offered by BaRISTA is beneficial for decoding visual stimuli as well. However, it will be very interesting to use the flexibility of spatial encoding enabled by our model for hypothesis-driven testing of spatial encoding scales on a variety of downstream tasks that exhibit different degrees of complexity in the future – including simpler sensory tasks. We will include discussion of this point as a possible future research direction in our manuscript.
>
> > Please clarify earlier on in your intro the working definition of neural tokenization.
>
> We thank the reviewer for their recommendation. We will revise the manuscript accordingly.
>
> **References:**
>
> [1] Christopher Wang et al. Brain Treebank: Large-scale intracranial recordings from naturalistic language stimuli.

---

> > ### Author Response · Authors · 2025-08-08
> >
> > Dear Reviewer 54tZ,
> >
> > As today is the last day of the author-reviewer discussion period, we would be very grateful if you could let us know if you have any feedback regarding our detailed rebuttal or any other further questions. We would be happy to provide additional clarifications that can help with your final re-assessment.
> >
> > Thank you again for your time and consideration.

---

### Official Review · Reviewer_xW3g · 2025-07-03

**Clarity:** 3
**Significance:** 3
**Originality:** 3
**Rating:** 5
**Confidence:** 4

**Summary:**

This paper introduces BaRISTA, an architecture that learns to interpret brain activity from iEEG recordings, using both spatial and temporal characteristics of the data. The authors introduce a masked latent reconstruction task to pre-train their model. They then investigate the performance impact of different spatial encodings, evaluating on various downstream tasks. The paper considers three distinct spatial scales, which are (from finest to coarsest): channel, atlas parcellation, and lobe. For the two coarser scales, the spatial encodings of channels within the same parcel or lobe are clustered together. The proposed architecture contains a dilated-convolution tokenizer, followed by a transformer-based component, whose outputs are fed into a downstream decoder. In the masked reconstruction task, the model receives a partially hidden input that it attempts to complete. The evaluation is performed on classification tasks, as well as a per-channel reconstruction task, with publicly available data.

**Questions:**

The authors are encouraged to consider the following items:
- Defending the claim that different scales are beneficial: The first action item is to attempt retraining or fine-tuning baselines (wherever possible) under different scales. The second is to complete Tab. 1 and Tab. 3 of the main paper with the missing scale combinations for BaRISTA. If any of the above are not applicable, please explain why that is so.
- Motivating and ablating the design choices of Sect. 3.2: Particular emphasis should be given to the S-vector and the attention encoder module. This is a core design component in the pipeline that requires further understanding. The first action item is to replace the single attention module with separate attention modules for space and time, and then compare the performance. The second action item would be to explain any additional benefits that motivate this design (e.g., computational savings, if any). Motivating or ablating the tokenizer and readout head designs would also be helpful.
- Although there is some information presented in App. B of the supplement, a more explicit elaboration of how fairness in the baseline comparisons is ensured would be appropriate. The total number of gradient updates for pre-trained and fine-tuned BaRISTA models should ideally be the same as that of the baselines. The action item is to present accurate counts of training steps taken for each compared system, including a breakdown of how many steps were taken during pre-training versus fine-tuning.

**Ethical Concerns:**

["NO or VERY MINOR ethics concerns only"]

**Final Justification:**

The authors addressed all points raised.

**Limitations:**

none identified

**Quality:**

3

**Strengths And Weaknesses:**

Strengths:
- The paper's core contributions, as presented, are technically sound.
- The proposed tasks are well-defined, and the architecture follows established methods in ML.
- The theoretical analysis is sufficient for understanding the proposed ideas.
- The supplemental materials provide many clarifying details and results (especially Tab. 3, Tab. 6, and Fig. 6).

Weaknesses:
- In processing the signal, the proposed pipeline essentially performs geometric clustering of channels based on a pre-defined atlas, and by using different atlases of varying granularity, we obtain different scales for the system. However, these clusterings rely on handcrafted models that, in turn, depend solely on anatomical structure. No concrete reasons are provided why such clusterings could not be learned directly, e.g., in the token space. This is partially addressed in Sect. 5, where [11] is cited as an approach for functional clustering. Given a downstream task, the question of where on the spectrum of possible clusterings lies the optimal (and how to find it), remains open.
- The use of a neurobiological prior (atlases) suggests that channel clustering may be universally beneficial for the examined tasks. If that is the case, the baselines' performance might show a similar improving trend when re-optimized under the same training protocol. However, since they are only trained on channel data and are not retrained at different scales, it is difficult to make a comparison.
- Several ideas introduced in Sect. 3.2 are not clearly motivated. For the encoder, it is not immediately obvious why a dilated CNN module is the best approach to capture continuous and/or oscillating features; several other options could be considered here. Similarly, the decoder being a linear layer is one possible choice, but could be ablated against differently sized MLPs, or even recurrent networks. Finally, and perhaps most crucially, there is the interleaved S vector and its associated attention mechanism. It is unclear what the benefit is of having a single attention module for space and time, as opposed to separate modules, and how it compares to other attention paradigms used in previous literature.

---

> ### Author Rebuttal · Authors · 2025-07-31
>
> We thank the reviewer for their thorough comments, recommendations, and excellent questions. We reply inline below.
>
> > retrain or fine-tune baselines under different scales
>
> This is an interesting point. First, we clarify that there was no notion of spatial encoding based on metadata (e.g., channel coordinates) in the iteration of Brant against which we compare [1]; vanilla index-based positional encoding [2] was used. Introducing spatial encoding to Brant would be adding new functionality and would require validation against the original pretrained model. The fairest way to perform such a comparison and validate the new functionality would be to pretrain on the same dataset the existing model was pretrained on. However we did not have access to that dataset nor did we have access to their pretraining code. Thus retraining/finetuning/validating Brant with different spatial scales would be outside of the scope of this work. The request to pretrain PopT with different spatial scales is possible, as we have the dataset the model was originally pretrained on and their code. We tried pretraining PopT with region-level encoding by replacing the $(x,y,z)$ encoding used in the original work with parcel-level spatial encoding. Using the default pretraining settings available on PopT's GitHub, we were unable to successfully pretrain the model: performance on downstream tasks was chance (~0.5 AUC on average). We suspect that this result may be due to the difference in pretraining tasks (we use a masked reconstruction task while PopT was trained on discriminative tasks). In order to train PopT to support different spatial scales, the pretraining task and/or optimization configurations would most likely need to be modified. Considering this result, we will revise the manuscript to clarify that the impact of spatial scales was seen for masked reconstruction pretraining, and that more work needs to be done to investigate the impact of spatial encoding in other pretraining tasks (e.g., discriminative tasks).
>
> > complete Tab 1 & Tab 3 of the main paper
>
> We clarify that Tab 2 is the complete version of Tab 1. We chose to present the comparisons against baseline models separately from the full 3x3 scale combinations in Tab 2 for the sake of readability; we will revise the text to make the relationship between Tab 1 and 2 clearer. Below we present the results of completed Tab 3 which we will add to the manuscript. We will revise section 4.4 to reflect these more comprehensive results, specifically noting that the granularity of spatial encoding may impact the model’s ability to learn individual channel properties, as evidenced by the drop in channel reconstruction performance with the lobe-level spatial encoding model.
>
> |Model|MSE $\downarrow$|$R^2 \uparrow$|
> |-|-|-|
> |Channels/Chans|0.397 &plusmn; 0.040|0.603 &plusmn; 0.040|
> |Chans/Parcels|0.354 &plusmn; 0.032|0.646 &plusmn; 0.032|
> |Chans/Lobes|0.478 &plusmn; 0.036|0.522 &plusmn; 0.036|
> |Parcels/Chans|0.391 &plusmn; 0.019|0.609 &plusmn; 0.019|
> |Parcels/Parcels|0.413 &plusmn; 0.023| 0.587 &plusmn; 0.023|
> |Parcels/Parcels|0.417 &plusmn; 0.027|0.583 &plusmn; 0.027|
> |Lobes/Chans|0.753 &plusmn; 0.039|0.247 &plusmn; 0.039|
> |Lobes/Parcels|0.951 &plusmn; 0.014|0.049 &plusmn; 0.014|
> |Lobes/Lobes|0.853 &plusmn; 0.029|0.147 &plusmn; 0.029|
> |Random init (chans encoding)|0.566 &plusmn; 0.028|0.434 &plusmn; 0.028|
> |Rand init (parcel enc)|0.846  &plusmn; 0.028|0.155  &plusmn; 0.028|
> |Rand init (lobe enc)|0.965 &plusmn; 0.022|0.035 &plusmn; 0.022|
>
> > Motivate & ablating the design choices of Sect. 3.2; explain additional benefits
>
> The choice to use a dilated CNN was based on prior works modeling uni/multivariate time-series activity [3-5]. We have now also performed an ablation on the temporal encoder choice and present results comparing against a linear projection (linear layer the size of our patch length $L$, similar to [1]) and a single layer univariate CNN with kernel size 5. For all models we used the best-performing encoding/masking pair presented in Tab 1 (parcels/channels) and report average AUC over 3 pretraining & 5 finetuning seeds (as in Tab 7). We will complete a more comprehensive ablation (use all spatial encoding/masking scales) and add it to the supplementary materials. Our results show that dilated CNN encoder achieves higher performance. With respect to computational savings: a dilated CNN is slower to train than a fully connected linear model but requires fewer parameters (302K for the linear projection vs 71.8K for the dilated CNN in our implementation). An RNN or SSM would have also been suitable alternatives but the training time for RNNs/SSMs is longer than that of a CNN. Since we wanted to pretrain several encoding/masking configurations in our analysis, we chose to optimize running time for the sake of experimentation and therefore opted for a CNN over a RNN/SSM. We will include this discussion on to computation savings/design choices in the manuscript.
>
> |Encoder (Parcels/Chans)|Sentence Onset|Speech|
> |-|-|-|
> |Linear projection|0.776  &plusmn; 0.01|0.763 &plusmn; 0.01|
> |Single layer univariate CNN (kernel=5)|0.749 &plusmn; 0.013|0.752 &plusmn; 0.012|
> |Dilated CNN (default)|**0.836 &plusmn; 0.010**|**0.847 &plusmn; 0.01**|
>
> We decided to use interleaved tokens (i.e., $\mathbf{S}$ vector) to give our model the capacity to learn spatiotemporal relationships between the channels within the same attention module (rather than each axis individually). We now show that this interleaved approach outperforms having separated attention modules through an ablation study in which we first pass our sequences through a temporal attention module (i.e., self-attention on the patches within each channel independently) and then through a spatial attention module (i.e., self-attention on the channels within each patch), similar to [1,6,7]. For fairness of comparison, we split our 12-layer transformer into 2 6-layer transformers each with 4 attention heads and the same hidden dimension 64. We used the best performing encoding/masking pair in Tab 1, but will add a more comprehensive analysis with all pairs in the supp materials. AUC scores are averaged across 3 pretraining & 5 finetuning seeds for each of the 7 test sessions. Our results show that the combined attention module achieves higher downstream performance.
>
> |Model|Sentence Onset|Speech|
> |-|-|-|
> |Interleaved $\mathbf{S}$|**0.862 &plusmn; 0.016**|**0.869 &plusmn; 0.016**|
> |Separate attention modules|0.828 &plusmn; 010|0.825 &plusmn; 0.011|
>
> We use a lightweight linear decoder to evaluate the quality of the learned embeddings/representations, as is common practice [8,9]. We were concerned that a more complex downstream decoder would have boosted the downstream performance regardless of the embedding/representation quality. We will clarify our choice of decoder in the text.
>
> > counts of training steps taken
>
> We thank the reviewer for the recommendation. For pretraining our model had 19,500 update steps, PopT pretraining involved 500,000 update steps, and Brant had 750,000 update steps. Because we ran training for a fixed number of epochs (as reported in Supp D.2), the total number of finetuning update steps was dependent on the downstream task and session (i.e., the number of training segments available). The average number of updates for BaRISTA across 7 test sessions and 2 downstream tasks was 252 (range 393 steps), for PopT it was 629 (range 982), and for Brant it was 1258 (range 1964). We chose the larger number of update steps for the baseline models to ensure they converged because we wanted to validate our model against their best performance. We trained Brant the longest as its finetuning learning rate was 1e3 times smaller than BaRISTA’s and 1e2 times smaller than that of PopT’s; we note that the LRs used reflect the rates used by the authors in the original works. We will include these numbers in the revised manuscript.
>
> > No concrete reasons are provided why such clusterings could not be learned directly…question of where on the spectrum of possible clusterings lies the optimal and how to find it remains open
>
> We agree with the reviewer that future work can explore using clusterings that are learned in a data-driven way, e.g., based on functional relationships or signal correlations. However, the use of handcrafted features in this work does not necessarily reflect a limitation in our model. We designed our model (1) to support both hand-crafted and data-driven (with minor code modifications) channel clusterings, and (2) to enable hypothesis testing by allowing users the flexibility to specify specific clustering labels based on side meta-information, which may not come from the data at hand but instead reflect domain knowledge (e.g., brain anatomy) and/or information from other modalities (e.g., structural images). For example, by utilizing clusters based on brain regions or functional relationships, users of our model can assess the significance and/or involvement of different regions or functional brain networks on various downstream tasks of interest. We will further clarify these points in the Discussion.
>
> **References**
>
> [1] D. Zhang, et al. Brant: Foundation Model for Intracranial Neural Signal
>
> [2] A. Vaswani et al. Attention Is All You Need
>
> [3] A. Oord, et al.. WaveNet: A Generative Model for Raw Audio
>
> [4] Z. Yue, et al. TS2Vec: Towards Universal Representation of Time Series
>
> [5] S. Bai, et al. An Empirical Evaluation of Generic Convolutional and Recurrent Networks for Sequence Modeling
>
> [6] G. Mentzelopoulos et al. Neural decoding from stereotactic EEG: accounting for electrode variability across subjects
>
> [7] G. Chau et al. Population Transformer: Learning Population-level Representations of Neural Activity
>
> [8] F. Pei et al. Neural Latents Benchmark '21: Evaluating latent variable models of neural population activity
>
> [9] T. Chen et al. A Simple Framework for Contrastive Learning of Visual Representations

---

> > ### Comment · Reviewer_xW3g · 2025-08-05
> > **Thank you**
> >
> > Thank you for all your elaborate responses which will affect my decision towards a more positive score.

---

> > > ### Author Response · Authors · 2025-08-06
> > >
> > > We sincerely thank the reviewer again for all their constructive comments and are grateful that our responses will affect their decision towards a more positive score.

---

### Note · Authors · 2025-08-12

We thank all reviewers for the helpful comments that improved our work. We introduced BaRISTA to enable flexible incorporation of spatial encoding/masking scales in modeling human iEEG data. We are glad reviewers acknowledged the novelty & impact of our core contribution & provided the feedback highlighted below:

1. Reviewer uiXd noted spatial flexibility gives “insight into the spatial scale of encoding” & “could steer the field towards adopting a better positional encoding”. Per their comments we added 1) a new volume task & 2) spectral analysis of channel reconstruction. We are glad our responses raised reviewer “confidence in the quality and impact of [our] work” & are grateful to hear they will increase their score and recommend acceptance.

2. Reviewer xW3g found our “core contributions” “technically sound”. Per their comments we 1) provided all spatial combinations for channel reconstruction & showed the benefits of 2) our temporal encoder & 3) interleaved space/time attention. We are glad to hear our “elaborate responses” will affect their decision to a more positive score.

3. Reviewer 54tZ found the spatial flexibility a “major achievement” saying that “varying spatial scales” is “very important when capturing high-level cognitive processes”. Per their comments we showed similar trends on a new visual optical flow task.

4. Reviewer eXGi noted “explor[ing] the impact of different spatial encoding/masking” and showing “spatial encoding at larger than channel-level encoding improves downstream decoding“ as a strength. The reviewer raised questions about a baseline, PopT, to which we: 1) clarified that our non-overlapping neural segments (Tab 1) were distinct from the overlapping segments used in PopT’s paper and prevented train/test information leakage as train/test splits are randomly generated from the segments; 2) reproduced the results in PopT’s paper with the original overlapping segments; 3) showed Tab 1 trends held with the reviewer’s suggested 80/10/10 k-fold chronological train/val/test splits; 4) got confirmation from the reviewer that despite their initial concern about “limited supervision samples” our number of samples “notably exceeds the 500” needed in PopT Fig 4 to get close to full performance; 5) added 2 tasks (volume, optical flow). While we did not get feedback from the reviewer after clarifying our new k-fold evaluation, we hope the analyses above addressed all concerns.

We again thank all reviewers for the helpful comments.

---

### Decision · Program_Chairs · 2025-09-17

**Decision:**

Accept (poster)

**Comment:**

This paper introduces BaRISTA, a JEPA-style pretraining framework for iEEG that flexibly incorporates spatial encoding at multiple scales (channel, parcel, lobe). Using a masked latent reconstruction objective, the authors systematically assess how spatial encoding/masking scales influence downstream decoding performance. Experiments on the Brain Treebank dataset show that encoding at parcel-level yields stronger results than channel-level encoding, while still preserving fine-grained information, and that the framework scales effectively to additional tasks such as speech/non-speech discrimination and sentence onset detection.

The strengths of this submission are clear. Reviewers appreciated the technical soundness and novelty of enabling flexible spatial scales in neurofoundation modeling (uiXd, 54tZ), with one noting this could “steer the field towards better positional encoding schemes for sEEG.” The paper is well written, supported by comprehensive ablations, and situates its contributions clearly. Reviewers also valued the empirical insight that larger spatial scales can improve high-level cognitive decoding while smaller scales preserve low-level reconstruction fidelity.

Some weaknesses remain. Evaluation was limited to a small set of tasks, and the PopT baseline raised concerns (eXGi). However, the rebuttal included additional experiments (e.g., K-fold evaluations, new volume and optical flow decoding tasks) and clarified baseline differences, which alleviated most concerns.

Overall, this is a technically solid and insightful contribution that advances understanding of spatial encoding in neural signal modeling. I recommend acceptance.